# PIXIU: A Large Language Model, Instruction Data and Evaluation Benchmark for Finance

**Qianqian Xie**
School of Computer Science
Wuhan University
Wuhan, Hubei, China
xieq@whu.edu.cn

**Weiguang Han**
School of Computer Science
Wuhan University
Wuhan, Hubei, China
han.wei.guang@whu.edu.cn

**Xiao Zhang**
Sun Yat-Sen University
Shenzhen, Guangdong, China
zhangx767@mail2.sysu.edu.cn

**Yanzhao Lai**
School of Economics and Management
Southwest Jiaotong University
Chengdu, Sichuan, China
laiyanzhao@swjtu.edu.cn

**Min Peng**
School of Computer Science
Wuhan University
Wuhan, Hubei, China
pengm@whu.edu.cn

**Alejandro Lopez-Lira**
University of Florida
alejandro.lopez-lira@warrington.ufl.edu

**Jimin Huang**
ChanceFocus AMC.
Shanghai, China
jimin@chancefocus.com

## Abstract

Although large language models (LLMs) have shown great performance in natural language processing (NLP) in the financial domain, there are no publicly available financially tailored LLMs, instruction tuning datasets, and evaluation benchmarks, which is critical for continually pushing forward the open-source development of financial artificial intelligence (AI). This paper introduces PIXIU, a comprehensive framework including the first financial LLM based on fine-tuning LLaMA with instruction data, the first instruction data with 128K data samples to support the fine-tuning, and an evaluation benchmark with 8 tasks and 15 datasets. We first construct the large-scale multi-task instruction data considering a variety of financial tasks, financial document types, and financial data modalities. We then propose a financial LLM called FinMA by fine-tuning LLaMA with the constructed dataset to be able to follow instructions for various financial tasks. To support the evaluation of financial LLMs, we propose a standardized benchmark that covers a set of critical financial tasks, including six financial NLP tasks and two financial prediction tasks. With this benchmark, we conduct a detailed analysis of FinMA and several existing LLMs, uncovering their strengths and weaknesses in handling critical financial tasks. The model, datasets, benchmark, and experimental results are open-sourced [1] to facilitate future research in financial AI.

---

[1] https://github.com/chancefocus/PIXIU

37th Conference on Neural Information Processing Systems (NeurIPS 2023) Track on Datasets and Benchmarks.

# 1 Introduction

Financial technology (FinTech) has been continually advanced by the development of natural language processing (NLP) and machine learning (ML) techniques, unlocking diversity capabilities from predicting stock price movements to advanced financial analytics (Araci, 2019; Han et al., 2023; Xie et al., 2023; Lopez-Lira and Tang, 2023; Li et al., 2023). Specifically, the most recent large language models (LLMs) (Brown et al., 2020)[2] have exhibited remarkable abilities in natural language understanding (NLU) and performing various tasks by following natural language instructions without training data. Despite these successes, the highly technical nature of financial texts requires domain-specific LLMs to understand complex financial language and concepts effectively. Such efforts include existing financial pre-trained language models (PLMs) such as finBERT (Araci, 2019), FinBERT (Yang et al., 2020) and FLANG (Shah et al., 2022). However, those models are considered small since their parameter size is below one billion, limiting their generalization ability. Recently, a proprietary financial LLM called BloombergGPT (Wu et al., 2023) with 50 billion parameters has been proposed by pre-training a Bloom-style LLM (Scao et al., 2022) on large-scale financial data.

Despite these efforts Liu et al. (2023), there remain several issues, as shown in Table 1. Firstly, BloombergGPT and its training data are not openly released. Currently, there are no open-sourced financial LLMs, which can hinder development in the research community. Secondly, previous financial PLMs and the latest BloombergGPT are not fine-tuned for following natural language instructions (also known as instruction tuning), which is critical for improving the zero-shot ability on dealing with downstream financial tasks (Wei et al., 2021; Ouyang et al., 2022). Thirdly, there are also no financial instruction data for supporting the instruction tuning of LLMs and evaluation benchmarks for comprehensively assessing and comparing the abilities of LLMs for financial tasks. We are thus motivated to consider the following research questions: 1) how can we develop efficient and openly available LLMs tailored for finance? 2) how can we build large-scale and high-quality financial instruction data? 3) how can we build the holistic financial evaluation benchmark for assessing financial LLMs?

Table 1: The comparison of pre-trained language models and large language models for finance. "Instruct" means whether the model can follow instructions. "NLP" and "Fin" mean if the model is evaluated with financial NLP tasks and financial prediction tasks.

| Model | Backbone | Size | Open Source | | Instruct | Language | Evaluation | | Release Date |
| | | | Model | Data | | | NLP | Fin | |
|---|---|---|---|---|---|---|---|---|---|
| finBERT (Araci, 2019) | BERT | 110M | ✓ | ✓ | ✗ | English | ✓ | ✗ | 08/27/19 |
| FinBERT (Yang et al., 2020) | BERT | 110M | ✓ | ✗ | ✗ | English | ✓ | ✗ | 06/15/20 |
| Mengzi-fin (Zhang et al., 2021) | RoBERTa | 103M | ✓ | ✗ | ✗ | Chinese | ✓ | ✗ | 10/13/21 |
| FLANG (Shah et al., 2022) | ELECTRA | 110M | ✓ | ✓ | ✗ | English | ✓ | ✗ | 10/31/22 |
| BBT-FinT5 (Lu et al., 2023) | T5 | 220M | ✓ | ✓ | ✗ | Chinese | ✓ | ✗ | 02/18/23 |
| BloombergGPT (Wu et al., 2023) | BLOOM | 50B | ✗ | ✗ | ✗ | English | ✓ | ✗ | 03/30/23 |
| FinMA | LLaMA | 7/30B | ✓ | ✓ | ✓ | English | ✓ | ✓ | 06/01/23 |

To deal with these research questions, we propose PIXIU (貔貅)[3], a comprehensive framework that includes the first open sourced fine-tuned financial LLM, FinMA, which is based on fine-tuning LLaMA (Touvron et al., 2023) with multi-task and multi-modal instruction data. Fig 1 presents an overview of multi-task and multi-modal instruction tuning of FinMA for diverse financial tasks. PIXIU also contains the first instruction data with 128K data samples to support the fine-tuning and a holistic evaluation benchmark with six financial NLP tasks and two financial prediction tasks. It has the following distinguishing features:

- **Open resources**. We have openly released the financial LLM, instruction tuning data, and datasets included in the evaluation benchmark, and implementation, to encourage open research and transparency in the research field.

- **Multi-task**. PIXIU includes multi-task instruction tuning data covering a diverse set of financial tasks, including six financial NLP tasks and two financial prediction tasks. The

---

[2] https://openai.com/blog/chatgpt

[3] PIXIU (貔貅) https://en.wikipedia.org/wiki/Pixiu is a mythical creature in Chinese folklore. It has the head of a dragon and the body of a lion and is believed to be an auspicious creature attracting money and good fortune.

multi-task instruction tuning has been proven to be critical for improving the model's generalization ability (Sanh et al., 2022; Longpre et al., 2023) to new tasks.

- **Multi-modality**. Our instruction tuning data consists of multi-modality financial data such as tables in financial reports and historical stock prices as time-series data for the stock-movement prediction tasks beyond texts. Moreover, they encompass diverse types of financial texts, including reports, news articles, tweets, and regulatory filings.

- **Diversity**. Compared with the evaluation tasks used in BloombergGPT and existing FLUE benchmark (Shah et al., 2022), which mainly cover financial NLP tasks, our evaluation benchmark includes financial prediction tasks such as stock movement prediction and credit scoring. It requires the model to fully exploit both natural texts and time-series data to extract essential information for accurate prediction. Compared with financial NLP tasks, the financial prediction task is more aligned with real-world scenarios and more challenging.

To build the multi-task and multi-modal instruction data, we collect open-released training data from diverse tasks, including financial sentiment analysis, news headline classification, named entity recognition (NER), question answering, text summarization, stock movement prediction, credit scoring and hawkish-dovish classification, and propose the diverse task-specific instructions written by domain experts for each task. We create a large-scale **f**inancial **i**nstruction **t**uning data (FIT) by assembling the task-specific instructions with data samples from each task. We thus propose the domain-specific LLM **FinMA** by conducting the multi-task instruction tuning on LLaMA with the building dataset. To evaluate our model and other LLMs holistically, we build the **F**inancial **L**anguage **U**nderstanding **A**nd **PR**ediction **E**valuation Benchmark (FLARE) covering 6 financial NLP tasks with 10 datasets, and 2 financial prediction task with 5 datasets.

Based on FLARE, we evaluate the performance of our model, BloombergGPT, and advanced LLMs in the general domain, such as ChatGPT[4] and GPT-4 (OpenAI, 2023). Experimental results show that: 1) FinMA significantly outperforms LLMs, including BloombergGPT, ChatGPT, and GPT-4 on most tasks in FLARE, including financial sentiment analysis, news headline classification, and stock movement prediction. This demonstrates the importance of tailoring the LLMs specifically for the financial domain. 2) Despite promising results on most tasks, FinMA underperforms BloombergGPT, ChatGPT, and GPT-4 on the question answering, which assesses the quantitative reasoning ability of LLMs. Our analysis finds that this is caused by the limitation of LLaMA on quantitative reasoning and mathematics. FinMA also shows limited performance on NER tasks although it outperforms BloomerbergGPT, which is also due to the drawbacks of LLaMA. 3) Compared with NLP tasks, all LLMs, including FinMA, ChatGPT and GPT-4, still present limited performance on stock movement prediction, indicating room for further improvement. 4) FinMA fine-tuned with both NLP and financial prediction tasks, presents the best performance on one of the stock prediction datasets, indicating the potential of task-specific instruction tuning of LLMs on financial prediction tasks.

Our contributions can be summarized as follows: 1) We introduce FIT, the first multi-task and multi-modal instruction tuning data in the financial domain, covering 5 tasks and 9 datasets with 128,640 (128K) data samples. 2) We introduce FLARE, the first evaluation benchmark with both financial natural language understanding and prediction tasks. 3) We introduce FinMA, the first openly released and instruction-following financial large language model, which achieves SOTA on 6 financial NLP tasks and 2 financial prediction tasks. 4) We compare FinMA and existing LLMs on FLARE. The results demonstrate the superiority of FinMA, the key limitations of LLMs for finance, and future directions to advance LLMs for finance.

## 2 Related Work

**Financial Language Models** Many PLMs for the financial domain have been proposed by continual pre-training PLMs with large-scale financial texts. Araci (2019) proposed the first financial PLM called finBERT that pre-trained BERT (Devlin et al., 2019) with open released financial corpus such as TRC2-financial[5] and Financial Phrase Bank (Malo et al., 2014). finBERT outperforms neural network methods such as LSTM in financial sentiment classification tasks. Yang et al. (2020) further proposed FinBERT by pre-training BERT with a 4.9 billion tokens financial communication

---

[4]https://openai.com/blog/chatgpt
[5]https://trec.nist.gov/data/reuters/reuters.html

corpus, which outperforms BERT on three financial sentiment classification datasets. Shah et al. (2022) proposed FLANG, a financial PLM with BERT and ELECTRA (Clark et al., 2020) as the backbone. Besides English, financial PLMs in other languages, such as Chinese, were also proposed, such as Mengzi-fin (Zhang et al., 2021) and BBT-FinT5 (Lu et al., 2023). Latest, Wu et al. (2023) proposed BloombergGPT, the first financial large language model with 50 billion parameters, that is pre-trained with mixed datasets from the general and financial domain. However, neither the model nor pre-trained domain datasets are not released. The model is also not instruction-following like other LLMs such as ChatGPT and GPT-4.

**Financial Evaluation Benchmark** Shah et al. (2022) proposed the first heterogeneous evaluation benchmark FLUE with 5 financial NLP tasks, including financial sentiment analysis (Malo et al., 2014), news headline classification (Sinha and Khandait, 2021), named entity recognition (Alvarado et al., 2015), structure boundary detection [6] and question answering (Maia et al., 2018). Lu et al. (2023) proposed the first Chinese financial evaluation benchmark BBT-CFLEB [7] with financial news classification, summarization, relation extraction, question answering, and negative news determination task, as well as sentiment classification task of financial social media texts. However, these benchmarks only consider financial NLP tasks and don't include financial prediction tasks, such as stock movement prediction, that are critical for evaluating the model's performance applied to real-world scenarios.

**Open Sourced Large Language Models** Recent studies have made efforts on democratic AI, where the representative work is LLaMA (Touvron et al., 2023) from Meta AI, an open-source LLM with parameters ranging from 7B and 13B to 65B. LLaMA-13B has comparable and even better performance than GPT-3 (Brown et al., 2020) with 175B parameters on common sense reasoning tasks. Following efforts have been proposed to improve LLaMA for instruction following like ChatGPT, by instruction tuning. Such as Taori et al. (2023) proposed Alpaca by fine-tuning LLaMA-7B with 52K instruction-following samples generated with the self-instruct method (Wang et al., 2022). Chiang et al. (2023) proposed Vicuna-13B by fine-tuning LLaMA-13B with 70K conversation data from ShareGPT [8]. It can generate better answers to user's questions compared with Alpaca. However, there are no open-sourced LLMs and instruction-tuning data focused on the financial domain.

## 3 FIT: Financial Instruction Tuning Dataset

In this section, we introduce our financial instruction tuning dataset FIT, including the background of raw data, tasks in FIT, and the construction process based on raw data. Different from existing financial datasets, FIT is the first instruction-tuning dataset for finance LLMs and includes financial prediction tasks except for financial NLP tasks, which is fundamental for real-world financial applications.

### 3.1 Raw Data

Derived from real-world finance scenarios, we build our financial instruction tuning dataset FIT based on the open-sourced data of various financial NLP and prediction tasks. Compared with the self-instruct method (Wang et al., 2022) commonly used by existing LLMs such as Alpaca, we choose to build instruction tuning datasets from open-sourced datasets due to the following reasons: 1) the open-sourced datasets are usually annotated by domain experts, showing high quality, 2) it has very low cost and has no limitation on commercial use unlike datasets constructed from ChatGPT or GPT-4, 3) these open-sourced datasets cover a variety of text types such as news, reports and tweets, as well as multi-modalities including time series data, tables, and texts. The details[9] of the raw data and instruction data are shown in Table 2.

**Financial Sentiment Analysis.** Financial sentiment analysis task has long been a critical task in the financial domain (Araci, 2019; Yang et al., 2020), aiming to analyze the sentiment information of the input financial texts. Following existing benchmark FLUE (Shah et al., 2022), we use two datasets: the Financial Phrase Bank (FPB) dataset (Malo et al., 2014) and FiQA-SA (Maia et al., 2018). FPB includes English sentences from financial news and their sentiment label of positive, negative, or

---

[6]https://sites.google.com/nlg.csie.ntu.edu.tw/finweb2021/shared-task-finsbd-3

[7]https://bbt.ssymmetry.com/evaluation.html

[8]https://sharegpt.com

[9]For further details of the data split and pre-processing, please refer to Appendix.

Table 2: The details of the raw data and instruction data.

| Data | Task | Raw | Instruction | Data Types | Modalities | License |
|------|------|-----|-------------|------------|------------|---------|
| FPB | sentiment analysis | 4,845 | 48,450 | news | text | CC BY-SA 3.0 |
| FiQA-SA | sentiment analysis | 1,173 | 11,730 | news headlines,tweets | text | Public |
| Headlines | news headline classification | 11,412 | 11,412 | news headlines | text | CC BY-SA 3.0 |
| FOMC | hawkish-dovish classification | 496 | 496 | FOMC transcripts | text | CC BY-NC 4.0 |
| NER | named entity recognition | 609 | 6,090 | financial agreements | text | CC BY-SA 3.0 |
| FiNER-ORD | named entity recognition | 1,080 | 1,080 | news articles | text | CC BY-SA 3.0 |
| FinQA | question answering | 8,281 | 8,281 | earnings reports | text,table | MIT License |
| ConvFinQA | question answering | 3,458 | 3,458 | earnings reports | text,table | MIT License |
| ECTSum | text summarization | 495 | 495 | earning call transcipts | text | Public |
| EDTSum | text summarization | 2,000 | 2,000 | news articles | text | Public |
| BigData22 | stock movement prediction | 7,168 | 7,168 | tweets,historical prices | text,time series | Public |
| ACL18 | stock movement prediction | 27,080 | 27,080 | tweets,historical prices | text,time series | MIT License |
| CIKM18 | stock movement prediction | 4,971 | 4,971 | tweets,historical prices | text,time series | Public |
| German | credit scoring | 1,000 | 1,000 | credit records | table | CC BY 4.0 |
| Australia | credit scoring | 690 | 690 | credit records | table | CC BY 4.0 |

neutral annotated by domain experts. FiQA-SA is another widely adopted dataset, which aims to predict the sentiment of English financial news and microblog posts on a scale of [-1,1], where 1 means the most positive.

**News Headline Classification.** The news headline classification task aims to analyze other information, such as price movement in financial texts. We use the Gold news headline dataset (Sinha and Khandait, 2021) consisting of news headlines from 2000 to 2019 about "gold" and their corresponding 9 tags: "price or not", "price up", "price down", "price stable", "past price", "future price", "past general", "future general", "asset comparison". The task is to conduct the binary classification for each tag of each data sample.

**Hawkish-dovish Classification.** The Hawkish-Dovish classification aims to classify sentences from monetary policy texts into a 'hawkish' or 'dovish' stance, unlike standard sentiment analysis. The key to this task lies in understanding the nuanced language of financial texts and being able to identify the economic implications conveyed through these 'hawkish' or 'dovish' signals. We use the FOMC dataset Shah et al. (2023a), including sentences extracted from the Federal Open Market Committee (FOMC) meetings, where each sentence is manually annotated as either 'hawkish' or 'dovish.

**Named Entity Recognition.** Named Entity Recognition (NER) task is to detect critical financial entities such as persons, organizations, and locations, which can be used to build financial knowledge graphs. We use two datasets: NER (Alvarado et al., 2015) and FiNER-ORD Shah et al. (2023b). includes sentences from public financial agreements through U.S. Security and Exchange Commission (SEC) filings, while FiNER-ORD consists of sentences from news articles. LOCATION (LOC), ORGANISATION (ORG) and PERSON (PER) entities of sentences from both datasets are manually annotated.

**Question Answering.** Question answering is the task of automatically answer a financial question based on the provided information. We use two datasets: FinQA (Chen et al., 2021) and ConvFinQA (Chen et al., 2022). FinQA consists of question-answering pairs annotated by experts and their corresponding earnings reports (including unstructured documents and tables) from S&P 500 companies. ConvFinQA is an expansion on FinQA that has conversations with the multi-turn question and answering over earnings reports.

**Text Summarization.** Text summarization aims to condense the long unstructured financial texts into the short summaries that capture crucial information and maintain factual consistency with the original long texts. We utilize two datasets: ECTSum Mukherjee et al. (2022) for extractive summarization and EDTSum Zhou et al. (2021) for abstractive summarization. ECTSum includes 2,425 long earnings call transcripts (ECT) and corresponding bullet-point summarization written by domain experts. EDTSum consists of financial news articles and corresponding titles as summaries.

**Stock Movement Prediction.** As one of the fundamental financial tasks, stock movement prediction has great potential value in real applications such as investment strategies. Following previous work (Soun et al., 2022), we frame the task as a binary classification problem, which is to predict the binary stock price movement given historical stock prices and tweets. If price movement is higher than 0.55%, it will be assigned to positive samples (1), or negative samples (-1) if it is lower than -0.5%. We adopt three commonly-used datasets: BigData22 (Soun et al., 2022), ACL18 (Xu and Cohen, 2018), and CIKM18 (Wu et al., 2018).

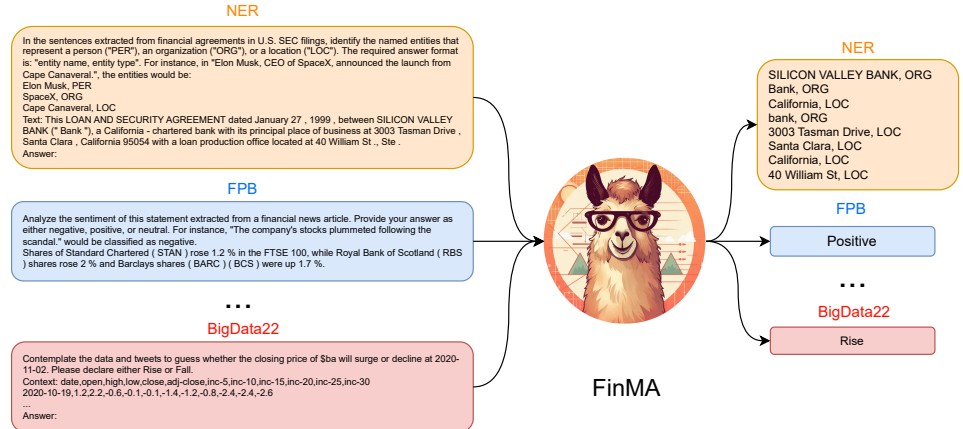

Figure 1: An overview of multi-task and multi-modal instruction tuning of FinMA for diverse financial tasks.

**Credit Scoring.** Credit scoring is a crucial task in financial services, aiming to classify consumers described by a set of attributes as either good or bad credit risks. We employ two datasets: German Credit Data (German) Hofmann (1994) and Australian Credit Data (Australian) Quinlan (1987). German contain 1000 instances of customers, each represented by 20 attributes, including status of existing checking account, credit history, etc., and the corresponding label of being a good or bad credit risk. Australian contains 690 instances with 14 attributes and the corresponding credit risk label. The task is to predict whether they are a good or bad credit risk based on these attributes.

## 3.2 Instruction construction

Base on the raw datasets, we further construct our financial instruction datasets, whose statistics are presented in Table 2. We ask domain experts to write 10 diverse instructions for all datasets except the ConvFinQA, where we only use one instruction. Since ConvFinQA is a multi-turn conversational question-answering dataset, which has diverse questions as instructions in nature. For BigData22, ACL18, CIKM18, we use the same instruction set, since they have the same data types of input data and task formulation[10]. Based on these prompts, we convert raw datasets from these tasks into instruction-tuning samples, by gathering human-designed instructions, and input texts along with responses of each dataset. For FPB, FiQA-SA, Headlines, NER, FiNER-ORD, ECTSum, EDTSum, German, Australian, FOMC, BigData22, ACL18, and CIKM18 datasets, we build instruction tuning samples with the following template:

> Instruction: [task prompt]    Text: [input text]    Response: [output]

[task prompt] is the prompt designed for each data, [input text] is the input financial data from each data, e.g. the historical prices and tweets for stock movement prediction datasets, [output] is the corresponding output for input text, e.g. sentiment label of input text from ["Positive", "Negative", "Neutral"] in FiQA-SA dataset. For FPB, FiQA-SA, and NER, we employ all 10 instructions for each sample, while we randomly sample one instruction for each sample in remaining datasets.

For FinQA and ConvFinQA, we use the following template:

> Instruction: [task prompt]    Context: [input context]    Question: [input question]    Response: [answer]

[input context] is the input contextual information for each data sample. For example, the input context can be filled with the text and table from the filling files for FinQA. ConvFinQA has multi-turn conversations with questions and answering. We transform each turn of the conversation for each data sample into one instruction via the template, which will append previous questions and answer in the [input context].

---

[10]The instruction examples are presented in Appendix

## 4    FinMA: Financial Large Language Model

We further build FinMA by fine-tuning LLaMA (Touvron et al., 2023) with FIT. We train four models: FinMA-7B and FinMA-30B by fine-tuning LLaMA 7B and 30B checkpoint with instruction tuning data covering NLP tasks, FinMA-7B-trade by fine-tuning LLaMA 7B checkpoint with instruction tuning data covering forecasting tasks, and FinMA-7B-full by fine-tuning LLaMA 7B with full instruction tuning data . We fine-tune LLaMA-7B and LLaMA-7B-trade with 15 epochs and LLaAM-7B-full with 3 epochs based on AdamW optimizer (Loshchilov and Hutter, 2017). The batch size is set to 32, the initial learning rate is 8e-6, and the weight decay is 1e-5. We also set warmup steps to 5% of all training steps. The maximum length of input texts is 2048. The FinMA-7B is fine-tuned on 8 A100 40GB GPUs. As for the FinMA-30B model, we fine-tune LLaMA-30B with 20 epochs, which is also based on the AdamW optimizer. The batch size is set to 24, the initial learning rate is 8e-6, the weight decay is 1e-5, and warmup steps to 5% of all training steps. The maximum length of input texts is 2048. Different from FinMA-7B, it can only be distributed fine-tuned on 128 A100 40GB GPUs.

## 5    FLARE: Financial Evaluation Benchmark

Based on FIT, we design our financial natural language understanding and prediction evaluation benchmark (FLARE). We randomly select validation sets from FIT to select the best model checkpoint, and test sets for evaluation. Compared with the existing benchmark FLUE (Sanh et al., 2022), FLARE covers financial prediction tasks in addition to NLP tasks[11]. We believe it is vital to include financial prediction tasks such as stock movement prediction, to comprehensively evaluate the performance of LLMs on the practical applications of the financial domain. We show the data statistics of validation, and test set for each dataset in Table 3. Following previous methods Li et al. (2023); Shah et al. (2022),

Table 3: The details of our evaluation datasets. To compare the performance with BloombergGPT whose test data is not openly released, we keep the same numbers and data distributions of our test datasets with that of BloombergGPT. To further evaluate the emergent and generalization ability of the LLMs, we only adopt the test data of FOMC, FINER-ORD, ECTSum, EDTSum, German, and Australian for evaluation on the FLARE. For ConvFinQA, we take each turn of the conversations as the instruction, whose number would be different from the number of conversations.

| Data | Task | Valid | Test | Evaluation |
|------|------|-------|------|------------|
| FPB (Malo et al., 2014) | sentiment analysis | 775 | 970 | F1, Accuracy |
| FiQA-SA (Maia et al., 2018) | sentiment analysis | 188 | 235 | F1 |
| Headlines (Sinha and Khandait, 2021) | news headline classification | 1,141 | 2,283 | Avg F1 |
| NER (Alvarado et al., 2015) | named entity recognition | 103 | 980 | Entity F1 |
| FiNER-ORD (Shah et al., 2023b) | named entity recognition | - | 1080 | Entity F1 |
| FinQA (Chen et al., 2021) | question answering | 883 | 1,147 | EM Accuracy |
| ConvFinQA (Chen et al., 2022) | question answering | 2,210 | 1,490 | EM Accuracy |
| BigData22 (Soun et al., 2022) | stock movement prediction | 798 | 1,470 | Accuracy, MCC |
| ACL18 (Xu and Cohen, 2018) | stock movement prediction | 2,560 | 3,720 | Accuracy, MCC |
| CIKM18 (Wu et al., 2018) | stock movement prediction | 431 | 1,140 | Accuracy, MCC |
| ECTSum Mukherjee et al. (2022) | text summarization | - | 495 | ROUGE, BERTScore, BARTScore |
| EDTSum Zhou et al. (2021) | text summarization | - | 2000 | ROUGE, BERTScore, BARTScore |
| German Hofmann (1994) | credit scoring | - | 1000 | F1, MCC |
| Australian Quinlan (1987) | credit scoring | - | 690 | F1, MCC |
| FOMC Shah et al. (2023a) | hawkish-dovish classification | - | 496 | F1, Accuracy |

we evaluate the performance of the sentiment classification task on FPB and FiQA-SA datasets, with the accuracy (ACC) and weighted F1 Score (F1). The performance of the news headline classification task is evaluated with the weighted averages of F1 score over all nine categories (Avg F1). For the performance of the NER task, we evaluate with the entity-level F1 score (Entity F1). The performance on the question-answering task is evaluated with exact match accuracy (EM Acc). For summarization tasks such as ECTSum and EDTSum, we assess the relevance and factuality of generated summaries with the ground truth using metrics such as ROUGE score Lin (2004), BERTScore Zhang et al. (2019), and BARTScore Yuan et al. (2021). As for the financial prediction task, following previous methods (Xu and Cohen, 2018; Xie et al., 2023), we evaluate the performance with the accuracy

---

[11]Following BloombergGPT, we don't include the structure boundary detection task included in FLUE because it is hard to be converted into the instruction following task.

(ACC) and the Matthews correlation coefficient (MCC) for stock price movement prediction, and the F1 score with MCC for credit scoring. Although Macro-F1 is more fair for the unbalanced dataset, in this paper we adopt the same metrics following previous methods to make a fair comparison.

## 6 Experiments on FLARE

The proposed FIT and FLARE allow to train, select the model, and evaluate the performance of LLMs on financial understanding and predictions. In this section, we investigate how powerful the FIT-fine-tuned FinMA and other LLMs are on FLARE. We compare FinMA with following LLMs: 1) BloombergGPT (Wu et al., 2023). The only large language model with 50B parameters pre-trained with the financial texts. 2) GPT-4 (OpenAI, 2023). A powerful instruction following large language model with around 1T parameters proposed by OpenAI. 3) ChatGPT. A instruction following large language model with 175B parameters from OpenAI. 4) Vicuna-7B (Zhang et al., 2022). An instruction following large language model by fine-tuning LLaMA-7B.

Following previous methods (Wu et al., 2023; Li et al., 2023), we report the 20-shot performance of BloombergGPT and the 5-shot performance of other baseline methods on the FIN dataset. We report the 5-shot performance of BloombergGPT on FPB and FiQA-SA. We report the 5-shot performance of all baselines on the Headlines dataset. For the remaining results, we report the zero-shot performance. The results of some baselines are based on human evaluations, since LLMs without fine-tuning will fail to generate answers pre-defined in the given instruction. All results of FinMA are conducted on zero-shot and can be automatically evaluated.

Table 4: The zero-shot and few-shot performance of different LLMs on the FLARE benchmark. Results of BloombergGPT, ChatGPT, and GPT4 on FPB, FiQASA, Headlines, NER, FinQA, and ConvFinQA are referenced from the paper (Li et al., 2023). The results of BloombergGPT are referenced from the original paper Wu et al. (2023). Test datasets were built to have the same data distribution with that of BloombergGPT and the performance of FinMA was directly compared with BloombergGPT following the previous method (Li et al., 2023). All results via our evaluations are the average of three runs. "-" represents the result that is currently unable to yield due to model size or availability, and "*" represents the result from the previous paper.

| Dataset | Metrics | Chat GPT | GPT 4 | Bloom berg GPT | Vicuna 7B | FinMA 7B | FinMA 7B-trade | FinMA 7B-full | FinMA 30B |
|---|---|---|---|---|---|---|---|---|---|
| FPB | F1 | 0.78* | 0.78* | 0.51* | 0.29 | **0.94** | 0.03 | **0.94** | 0.88 |
| | Acc | 0.78* | 0.76* | - | 0.26 | **0.94** | 0.12 | **0.94** | 0.87 |
| FiQA-SA | F1 | 0.60 | 0.80 | 0.75* | 0.32 | 0.85 | 0.16 | 0.82 | **0.87** |
| Headlines | AvgF1 | 0.77* | 0.86* | 0.82* | 0.60 | **0.97** | 0.28 | **0.97** | **0.97** |
| NER | EntityF1 | 0.77* | **0.83*** | 0.61* | 0.12 | 0.59 | 0.00 | 0.64 | 0.62 |
| FINER-ORD | EntityF1 | 0.28 | **0.77** | - | 0.00 | 0.00 | 0.00 | 0.00 | 0.00 |
| FinQA | EmAcc | 0.58* | **0.63*** | - | 0.00 | 0.06 | 0.00 | 0.04 | 0.11 |
| ConvFinQA | EmAcc | 0.60* | **0.76*** | 0.43* | 0.00 | 0.25 | 0.00 | 0.20 | 0.40 |
| BigData22 | Acc | 0.53 | **0.54** | - | 0.44 | 0.45 | 0.45 | 0.51 | 0.47 |
| | MCC | -0.025 | 0.03 | - | -0.05 | 0.02 | 0.00 | 0.02 | **0.04** |
| ACL18 | Acc | 0.50 | **0.52** | - | 0.50 | 0.49 | 0.49 | 0.51 | 0.49 |
| | MCC | 0.005 | 0.02 | - | 0.02 | -0.01 | **0.03** | 0.03 | 0.00 |
| CIKM18 | Acc | 0.55 | **0.57** | - | 0.44 | 0.43 | 0.43 | 0.50 | 0.43 |
| | MCC | 0.01 | 0.02 | - | -0.03 | -0.02 | -0.003 | **0.08** | -0.05 |
| EDTSUM | Rouge-1 | 0.17 | 0.2 | - | **0.22** | 0.09 | 0.05 | 0.13 | 0.17 |
| | Rouge-2 | 0.08 | 0.09 | - | **0.10** | 0.04 | 0.02 | 0.06 | 0.08 |
| | Rouge-N | 0.13 | 0.15 | - | **0.17** | 0.08 | 0.05 | 0.10 | 0.14 |
| | BertScore | 0.66 | **0.67** | - | 0.61 | 0.56 | 0.51 | 0.38 | 0.54 |
| | BartScore | -3.64 | **-3.62** | - | -4.13 | -6.12 | -6.91 | -5.71 | -5.24 |
| ECTSUM | Rouge-1 | 0.00 | 0.00 | - | 0.00 | 0.00 | 0.00 | 0.00 | 0.00 |
| | Rouge-2 | 0.00 | 0.00 | - | 0.00 | 0.00 | 0.00 | 0.00 | 0.00 |
| | Rouge-N | 0.00 | 0.00 | - | 0.00 | 0.00 | 0.00 | 0.00 | 0.00 |
| | BertScore | 0.00 | 0.00 | - | 0.00 | 0.00 | 0.00 | 0.00 | 0.00 |
| | BartScore | -5.18 | -5.18 | - | -5.18 | -5.18 | -5.18 | -5.18 | -5.18 |
| German | F1 | 0.20 | **0.55** | - | 0.52 | 0.17 | 0.52 | 0.17 | 0.53 |
| | MCC | -0.10 | -0.02 | - | **0.00** | 0.00 | -0.07 | 0.00 | -0.07 |
| Australian | F1 | 0.41 | **0.74** | - | 0.26 | 0.41 | 0.26 | 0.41 | 0.46 |
| | MCC | 0.00 | **0.47** | - | 0.00 | 0.00 | 0.00 | 0.00 | -0.01 |
| FOMC | F1 | 0.64 | **0.71** | - | 0.19 | 0.49 | 0.10 | 0.49 | 0.43 |
| | Acc | 0.6 | **0.69** | - | 0.28 | 0.47 | 0.25 | 0.46 | 0.53 |

## 6.1 Results

**Overall Performance.** For financial NLP tasks, as shown in Table 4[12], our fine-tuned model FinMA significantly outperform other LLMs on FPB, FiQA-SA and Headlines datasets, showing the importance of domain specific instruction tuning on improving the performance of LLMs in the specific domain. For example, FinMA-30B outperforms GPT-4 by 10% F1 score, and BloombergGPT by 37% F1 score on the FPB dataset. On the NER dataset, FinMA-7B also outperforms BloombergGPT and other LLMs, and achieve competitive results compared with ChatGPT and GPT-4. For FinQA and ConvFinQA which requires complex numeric reasoning, there is a large gap between the performance of GPT and FinMA. As reported in existing studies (Touvron et al., 2023; Lewkowycz et al., 2022), LLaMA includes no mathematical datasets for pre-training, resulted in poor performance on the mathematical benchmark datasets such as GSM8K (Cobbe et al., 2021). Our results are also consistent with previous studies which reveals that LLaMA with larger parameters present better performance on mathematical benchmark datasets. The performance of FinMA-30B is significantly better than that of FinMA-7B on FinQA and ConvFinQA. This finding indicate the importance of numeric reasoning for financial question answering, which could be the potential direction for advancing LLMs in the finance area. Despite the strengths exhibited by our methods on known tasks, they underperform relative to GPT-4 on unseen tasks such as FINER-ORD, EDTSUM, ECTSUM, and FOMC. This performance gap suggests that a more diverse set of domain-specific tasks is needed for effective fine-tuning. Specifically, in FINER-ORD and ECTSUM, we employ a complex prompt design that requires the model to generate label sequences directly. The results indicate that our fine-tuned models consistently fail to produce outputs in the desired formats. While models like ChatGPT and GPT-4 demonstrate some capability for token labeling in FINER-ORD, they also struggle to generate sentence labels for ECTSUM, particularly when faced with longer contextual information. In contrast to existing financial benchmarks, FLARE offers a more comprehensive suite of both generation and classification tasks, thereby providing a fuller assessment of Large Language Models' capabilities in the realm of financial NLP.

For financial prediction tasks, all LLMs including FinMA, ChatGPT and GPT-4 struggle in stock movement prediction as previous methods[13]. While FinMA-7B-trade has been fine-tuned specifically for the task of stock movement prediction, the observed performance gains in this area are marginal at best. After fine-tuned with both NLP and financial prediction tasks, FinMA-7B-full can achieve a significantly better performance on ACL18 dataset compared with ChatGPT and GPT-4. However, it still presents almost zero MCC on the other two datasets like ChatGPT and GPT-4. Similarly, all tested methods, with the exception of GPT-4, exhibited either zero or negative Matthews Correlation Coefficient (MCC) values on the credit scoring task. This highlights their limitations in accurately forecasting individual default risks. While GPT-4 shows a marked improvement in performance on the Australian dataset, it fails to register a positive MCC on the German dataset. Such limitations can be attributed to the challenges associated with the tabular data input and the highly imbalanced label distribution inherent to credit scoring tasks. This indicates the complexity and challenging of the financial prediction tasks in FLARE. Compared with existing financial benchmarks that focusing on NLP tasks, FLARE provides exciting opportunities for the improvement of LLMs on the fundamental of financial academic studies and applications. It also demonstrates the importance of multiple task learning in the financial domain for LLMs, which can provide essential domain knowledge and skills to handle complex applications in this area.

**Further Analysis.** We further analyze the influences of model size, and instruction tuning data on the performance of LLMs on different tasks. FinMA-30B has no significantly better performance than FinMA-7B on most NLP tasks and the stock movement prediction task. Apparently, the quality of the instructions rather than the model size is critical for the performance of these tasks. For the complex question-answering tasks such as ConvFinQA, as shown in Table 4, the larger LLaMA model generally has better performance. Particularly, Vicuna-7B based on LLaMA-7B has the worst performance, which are also consistent with previous findings (Cobbe et al., 2021) that LLaMA with larger parameters presents better performance on mathematical benchmark datasets. In contrast, for generation tasks such as abstractive summarization (EDTSUM), Vicuna-7B presents the best performance while fine-tuned models shows a decreased performance on almost all metrics. This may indicate that fine-tuning with only classification tasks could lead to better classification performance but also compromise the generation ability. For financial prediction tasks and NLP tasks that are not

---

[12]For the performance of other LLMs, please see Appendix.
[13]For the performance of traditional methods, please see Appendix.

included in the fine-tuning instruction dataset of FinMA-7B and FinMA-30B, i.e, FOMC, FINER-ORD, Australian, and ACL18, our models present limited improvement. While it demonstrates some degree of emerge ability, the performance gap compared with GPT-4 indicates the needs for further optimization. However, FinMA-7B-full fine-tuned with both NLP and prediction datasets, has shown significantly better performance on financial prediction datasets, and comparable performance on NLP tasks with FinMA-7B and GPT-4. This indicates the potential of LLMs to be further adapted and applied directly on financial prediction tasks via pre-training and fine-tuning on diversed domain datasets.

## 7 Limitations

Despite the positive contributions of this study, we recognize the following limitations: 1) **Model and Training Constraints**: We only present FinMA models up to 30B. Due to computational constraints, FinMA-30B has not been fine-tuned on the full dataset. 2) **Complex Task Performance**: FinMA, due to the limitation of the backbone model LLaMA, struggles with tasks requiring quantitative reasoning, such as financial question answering, and the difficult financial prediction task. 3)**Resource Constraints and Generalizability**: The development of FinMA, FIT, and FLARE is influenced by available resources and handcrafted instructions, potentially affecting model diversity and generalizability. The maximum input size of FinMA is also limited by the maximum input texts that can be handled by the backbone model LLaMA. 4)**Potential Negative Impacts**: While our study primarily focuses on the positive aspects and advancements of financial language understanding models, it is important to acknowledge the potential negative impacts associated with their use, such as the spread of financial misinformation or unethical market influence. We recommend using our method for academic research only.[14]

## 8 Conclusion

In this work, we presented PIXIU, encompassing the first open-sourced financial large language model FinMA, the instruction tuning dataset FIT, and the evaluation benchmark FLARE. Through extensive evaluation, we demonstrated the effectiveness of FinMA across various financial tasks, showing the potential of domain-specific instruction tuning of large language models in the financial domain. However, challenges such as improving performance on complex tasks and addressing resource constraints remain. Our open-source contribution aims to facilitate further research and innovation in financial language understanding, prediction, and LLMs, toward more useful and safe LLMs in the field of finance.

## Acknowledgement

The PIXIU project is mainly supported by ChanceFocus AMC and partially supported by the National Key Research and Development Program of China (No.2021ZD0113304) and the General Program of Natural Science Foundation of China (NSFC) (Grant No.62072346). Additional funding was provided by the Key R&D Project of Hubei Province (Grant No.2021BBA099, No.2021BBA029) and by the Joint & Laboratory on Credit Technology.

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

# A  Instructions

Table 5 presents all the prompts for each dataset in the FLARE benchmark and FIT instruction dataset.

Table 5: The example prompt for each dataset. FiQA-SA has two types of text, including news headlines and tweets. We will fill the detailed text type into {category} for each data sample. For stock movement prediction data such as BigData22, we will fill {tid} and {point} with the detailed stock name and time from each data sample.

| Data | Prompt |
|---|---|
| FPB | "Analyze the sentiment of this statement extracted from a financial news article.
Provide your answer as either negative, positive or neutral.
For instance, 'The company's stocks plummeted following the scandal.' would be classified as negative." |
| FiQA-SA | "What is the sentiment of the following financial {category}:
Positive, Negative, or Neutral?" |
| Headlines | "Consider whether the headline mentions the price of gold.
Is there a Price or Not in the gold commodity market indicated in the news headline?
Please answer Yes or No." |
| NER | "In the sentences extracted from financial agreements in U.S. SEC filings,
identify the named entities that represent a person ('PER'), an organization ('ORG'),
or a location ('LOC'). The required answer format is: 'entity name, entity type'.
For instance, in 'Elon Musk, CEO of SpaceX, announced the launch from Cape Canaveral.',
the entities would be: 'Elon Musk, PER; SpaceX, ORG; Cape Canaveral, LOC'" |
| FiNER-ORD | "In the list of tokens, identify 'Person', 'Location', and 'Organisation' and label each accordingly.
If the entity spans multiple tokens, use the prefix B-PER, B-LOC, or B-ORG for the first token,
and I-PER, I-LOC, or I-ORG for the subsequent tokens of that entity. The beginning of each separate
entity should always be labeled with a B-PER, B-LOC, or B-ORG prefix. If the token does not fit into
any of the three named categories, or is not a named entity, label it as 'O'. Each line should contain one
token and its corresponding label, separated by a colon. Do not combine tokens on your own. The format
for each line should be: 'token:label'. Text: And all because you failed to prepare ! Answer:" |
| FinQA | "Given the financial data and expert analysis, please answer this question:" |
| ConvFinQA | "In the context of this series of interconnected finance-related queries and the additional information
provided by the pretext, table data, and post text from a company's financial filings,
please provide a response to the final question. This may require extracting information
from the context and performing mathematical calculations. Please take into account the information provided in
the preceding questions and their answers when formulating your response:" |
| BigData22 | " Contemplate the data and tweets to guess whether the closing price of {tid} will surge or decline at {point}.
Please declare with either Rise or Fall." |
| ECTSum | "Given the following article, please produce a list of 0 and 1, each separated by ' ' to indicate which sentences
should be included in the final summary. The article's sentences have been split by ' '. Please mark each sentence
with 1 if it should be included in the summary and 0 if it should not." |
| EDTSum | "You are given a text that consists of multiple sentences. Your task is to perform abstractive summarization on this text. Use
your understanding of the content to express the main ideas and crucial details in a shorter, coherent, and natural sounding text." |
| German | "Assess the creditworthiness of a customer using the following table attributes for financial status. Respond with either
'good' or 'bad'. And the table attributes including 13 categorical attributes and 7 numerical attributes are as follows:" |
| FOMC | "Examine the excerpt from a central bank's release below. Classify it as HAWKISH if it advocates for a tightening
of monetary policy, DOVISH if it suggests an easing of monetary policy, or NEUTRAL if the stance is unbiased.
Your response should return only HAWKISH, DOVISH, or NEUTRAL." |

# B  Traditional Methods for Stock Movement Forecasting

In the context of stock movement prediction, traditional models, as summarized in Table 6, have long been employed but face significant challenges in achieving consistently high levels of accuracy. This underscores the inherent difficulty of the task at hand. In contrast, Large Language Models (LLMs) introduce a level of adaptability by learning from multiple tasks, although they also have limitations such as numeric understanding and reasoning. The task's difficulty is thus a common challenge for both traditional models and LLMs.

Table 6: Movement prediction performance of non-LLM models vs FinMA, measured with the accuracy (ACC) and the Matthews correlation coefficient (MCC). The best of non-LLM models is in red and the best of all is in bold.

| Method | BIGDATA22 | | ACL18 | | CIKM18 | |
|---|---|---|---|---|---|---|
| | ACC | MCC | ACC | MCC | ACC | MCC |
| Logistic regression (LR) | 0.53 | 0.02 | 0.52 | 0.04 | 0.53 | -0.04 |
| Random forest (RF) | 0.47 | -0.11 | 0.52 | 0.03 | 0.54 | 0.01 |
| LSTM | 0.51 | 0.01 | 0.53 | 0.06 | 0.53 | 0.02 |
| Attention LSTM (ALSTM) | 0.49 | -0.03 | 0.52 | 0.04 | 0.53 | -0.01 |
| Adv-ALSTM | 0.50 | 0.01 | 0.53 | 0.07 | 0.54 | 0.02 |
| DTML | 0.52 | 0.07 | 0.58 | 0.18 | 0.54 | -0.00 |
| XGBoost | 0.52 | -0.04 | 0.49 | -0.02 | **0.58** | 0.07 |
| XGBRefressor | 0.46 | -0.13 | 0.50 | -0.01 | 0.53 | -0.03 |
| ALSTM-W | 0.48 | -0.01 | 0.53 | 0.08 | 0.54 | 0.03 |
| ALSTM-D | 0.49 | 0.01 | 0.53 | 0.07 | 0.50 | -0.04 |
| StockNet | 0.53 | -0.00 | 0.54 | -0.03 | 0.52 | -0.02 |
| SLOT | **0.55** | **0.10** | **0.59** | **0.21** | 0.56 | **0.09** |
| FinMA-7B | 0.48 | 0.04 | 0.50 | 0.00 | 0.56 | -0.02 |
| FinMA-30B | 0.47 | 0.04 | 0.49 | 0.00 | 0.43 | -0.05 |
| FinMA-7B-full | 0.49 | 0.01 | 0.56 | 0.10 | 0.53 | -0.03 |

## C  Performance of BERT based Large Language Models

Table 7 presents previously reported performance of pre-trained language models (PLMs) including FinBERT and FLANG-BERT across three selected tasks under the FLARE benchmark. Although both models demonstrate impressive results in specific performance metrics, it's crucial to note that they have been pre-trained with large-scale financial texts and utilize task-specific headers for different tasks, which is the main reason for their better performance. However, compared with LLMs, these PLMs can't be adaptable to unseen tasks without supervised fine-tuning, thus having the bottleneck of adapting to a multi-task environment and zero-shot learning. For LLMs, our results suggest their performance still underperforms these finely tuned PLMs on some tasks, but offer greater flexibility, adaptability, and zero-shot ability. They have the advantage of learning directly from prompts and being more flexible in handling a wide array of tasks, even without the need for labeled training data. The underperformance of LLMs compared with PLMs also highlights the need for domain-specific pre-training of LLMs in the future, to further improve their performance in the specific domain.

These observations align with recent studies, such as Shah and Chava (2023), which also observed that zero-shot LLMs like ChatGPT offer respectable performance across financial tasks without the need for labeled data. The recent work by Ni et al. (2023), also highlights the challenges of fine-tuned PLMs adapting to unseen and multiple tasks.

Table 7: Results of BERT-based Encoder-Decoder Large Language Models (LLMs) across multiple tasks reported in previous papers.

| Method | FPB | Headline | NER |
|---|---|---|---|
| | Accuracy | AvgF1 | F1 |
| FinBERT | 0.872 | 0.968 | 0.8 |
| FLANG-BERT | 0.912 | 0.972 | 0.83 |

# D Performance of Previous Large Language Models

Table 8 provides a detailed comparison of zero-shot and few-shot performance metrics for various Large Language Models (LLMs)—GPT NeoX 20B, OPT 66B, and BLOOM—across multiple datasets. These metrics, obtained from a previous paper, serve as a valuable baseline for understanding the capabilities of these models in both zero-shot and few-shot scenarios under the FLARE benchmark.

Table 8: The zero-shot and few-shot performance of different LLMs on the FLARE benchmark. Results are referenced from previous paper.

| Dataset | Metrics | GPT NeoX 20B | OPT 66B | BLOOM |
|---|---|---|---|---|
| FPB | F1 | 0.45* | 0.49* | 0.50* |
| | Acc | 0.38 | - | - |
| FiQA-SA | F1 | 0.51* | 0.52* | 0.53* |
| Headlines | AvgF1 | 0.73* | 0.79* | 0.77* |
| NER | EntityF1 | 0.61* | 0.57* | 0.56* |
| FinQA | EmAcc | 0.00 | - | - |
| ConvFinQA | EmAcc | 0.28* | 0.30* | 0.36* |
| BigData22 | Acc | 0.41 | - | - |
| | MCC | 0.08 | - | - |
| ACL18 | Acc | 0.35 | - | - |
| | MCC | 0.00 | - | - |
| CIKM18 | Acc | 0.25 | - | - |
| | MCC | -0.12 | - | - |
| EDTSUM | Rouge-1 | 0.02 | - | - |
| | Rouge-2 | 0.01 | - | - |
| | Rouge-N | 0.02 | - | - |
| | BertScore | 0.48 | - | - |
| | BartScore | -5.72 | - | - |
| ECTSUM | Rouge-1 | 0.00 | - | - |
| | Rouge-2 | 0.00 | - | - |
| | Rouge-N | 0.00 | - | - |
| | BertScore | 0.00 | - | - |
| | BartScore | -5.18 | - | - |
| German | F1 | 0.17 | - | - |
| | MCC | 0.02 | - | - |
| Australian | F1 | 0.00 | - | - |
| | MCC | 0.00 | - | - |
| FOMC | F1 | 0.37 | - | - |
| | Acc | 0.27 | - | - |

