# A    Motivation For Datasheet Creation

## A.1    Why was the datasheet created? (e.g., was there a specific task in mind? was there a specific gap that needed to be filled?)

The PIXIU framework was created in response to the lack of open-source, financial-specific Large Language Models (LLMs), instruction-tuning datasets, and evaluation benchmarks. These resources are critical for advancing the field of financial artificial intelligence (AI). Existing LLMs were not adequately fine-tuned to handle natural language instructions related to financial tasks, thus limiting their performance in this crucial domain.

PIXIU, therefore, aims to fill this gap by offering a comprehensive set of resources tailored to the financial field. These include the first financial LLM (FinMA), a large instruction dataset comprising 128K samples, and an evaluation benchmark encompassing eight tasks and fifteen datasets. These resources were designed to improve the understanding of complex financial language and concepts, enhance the performance of AI in financial tasks, and foster the open-source development of financial AI.

## A.2    Has the dataset been used already? If so, where are the results so others can compare (e.g., links to published papers)?

No, this paper represents the first usage of the PIXIU dataset. The results obtained from using this dataset have been documented in the paper and include a detailed analysis of the performance of the FinMA model and several existing Large Language Models (LLMs) in handling critical financial tasks. As of now, the results are accessible within the paper itself, which is published on ArXiv() and accessible to the public. Furthermore, the performance results of models using the PIXIU framework, including FinMA, are also available on the project's GitHub page[1].

## A.3    What (other) tasks could the dataset be used for?

The PIXIU project is designed to be highly versatile and can be used for a wide range of financial tasks beyond the ones it was initially used for. It can serve as a resource for training and fine-tuning other financial Large Language Models (LLMs), potentially leading to improvements in various areas of financial technology (FinTech). These areas can include financial sentiment analysis, fraud detection, financial forecasting, personalized financial advice, and regulatory compliance.

## A.4    Who funded the creation dataset?

PIXIU is a collaborative project carried out by researchers from several institutions, including Wuhan University, Sun Yat-Sen University, Southwest Jiaotong University, and the University of Florida. The project is funded by ChanceFocus AMC.

## A.5    Any other comment?

PIXIU is a significant breakthrough in the domain of financial AI. It aims to bridge the gap in the lack of open-source financial LLMs and benchmarks for their evaluation. The introduction of FinMA, a large language model tailored specifically for financial tasks, is a key feature of PIXIU. It is fine-tuned to follow natural language instructions, enhancing its performance in downstream financial tasks.

Moreover, PIXIU is designed to foster transparency and collaboration in the field. It openly provides the financial LLM, instruction tuning data, and datasets included in the evaluation benchmark to

---

[1] https://github.com/chancefocus/PIXIU

encourage open research. It covers a diverse set of financial tasks, thus offering a versatile tool for both academic researchers and industry professionals.

By providing these resources and promoting open-source development, PIXIU is set to push forward the frontier of financial AI, offering a comprehensive framework for assessing financial LLMs and fostering a deeper understanding of complex financial language and concepts. This project is a significant step towards the integration of advanced AI techniques into the financial industry, paving the way for a wide range of applications such as predicting stock price movements and advanced financial analytics.

# B  Datasheet Composition

## B.1  What are the instances?(that is, examples; e.g., documents, images, people, countries) Are there multiple types of instances? (e.g., movies, users, ratings; people, interactions between them; nodes, edges)

The instances in the instruction dataset for the PIXIU model can be understood as the individual samples of data that are used to fine-tune the model. These instances are diverse and cover multiple types, as they are derived from various tasks and data types in the financial domain.

For example, in the task of named entity recognition, an instance would be a sentence extracted from a financial agreement, along with the instruction to identify named entities that represent a person, organization, or location.

The data types used in our instances include:

1. Textual Data: Many instructions refer to the processing of textual data, which includes reports, news articles, news headlines, regulatory filings, tweets, and financial agreements. Examples of tasks include sentiment analysis, news headline classification, and named entity recognition.

2. Time-Series Data: Certain instructions refer to time-series data, specifically stock price data, often in conjunction with textual data. This is primarily used in the stock movement prediction task.

3. Tabular Data: Some instructions involve the processing of tabular data, which are usually extracted from company's financial filings. This type of data is particularly relevant in question-answering tasks where the model needs to extract and utilize information from these tables.

## B.2  How many instances are there in total (of each type, if appropriate)?

The instruction dataset is broken down by each task type, with the total number of instances for each as follows:

- Sentiment Analysis: 60,180 instances

- News Headline Classification: 11,412 instances

- Named Entity Recognition: 6,090 instances

- Question Answering: 11,739 instances

- Stock Movement Prediction: 39,219 instances

This results in a total of 128,640 instances in the instruction dataset.

**B.3 What data does each instance consist of ? "Raw" data (e.g., unprocessed text or images)? Features/attributes? Is there a label/target associated with instances? If the instances related to people, are subpopulations identified (e.g., by age, gender, etc.) and what is their distribution?**

- **Sentiment Analysis:** Each instance consists of raw text data derived from news reports or tweets. The attributes include the text data and the associated sentiment label, which can be positive, negative, or neutral. No subpopulations are identified in this task.

- **News Headline Classification:** Each instance consists of raw text data in the form of news headlines. The attributes include the headline text and the associated label indicating whether or not the headline mentions the price of a commodity (Yes or No). No subpopulations are identified in this task.

- **Named Entity Recognition:** Instances comprise raw text from financial agreements and filings. The attributes include the text data and the named entities identified within it. Entities are labelled as person ('PER'), an organization ('ORG'), or a location ('LOC'). Subpopulations could potentially be inferred from the entities, but they are not directly identified.

- **Question Answering:** Each instance includes raw text data from company financial filings. The attributes include the question, the context (pretext, table data, post text), and the associated answer. No subpopulations are identified in this task.

- **Stock Movement Prediction:** Instances consist of raw text data from reports or social media posts and time-series data from stock prices. The attributes include the text data, the historical prices, and the label indicating the predicted stock movement (either Rise or Fall). Subpopulations could potentially be inferred from the companies under analysis, but they are not directly identified.

**B.4 Is there a label or target associated with each instance? If so, please provide a description.**

Each task in the dataset has a specific target or label associated with it, as described below:

- **Financial Sentiment Analysis:** The labels for the FPB and FiQA-SA datasets are sentiment categories, classified as positive (sentiment score in the range [0.1,1]), negative (sentiment score in the range [-1,-0.1)), or neutral (sentiment score in the range [-0.1, 0.1)). These labels are derived from the original sentiment scores provided in the datasets.

- **News Headline Classification:** The label for this task is binary, indicating whether the news headline mentions the price of a commodity or not. Therefore, the labels can be either 'Yes' or 'No'.

- **Named Entity Recognition:** For the NER dataset, the labels are categories of named entities found in the text. The 'MISCELLANEOUS' entities are discarded, and only those labeled as a person ('PER'), an organization ('ORG'), or a location ('LOC') are retained.

- **Question Answering:** In the FinQA and ConvFinQA datasets, the label is the correct answer to the question posed in the context of financial data. The answers may range from specific numeric values to complex text responses, depending on the nature of the question.

- **Stock Movement Prediction:** For stock prediction tasks in the BigData22, ACL18, and CIKM18 datasets, the label is the predicted direction of stock movement: 'Rise' or 'Fall'.

**B.5 Is any information missing from individual instances? If so, please provide a description, explaining why this information is missing (e.g., because it was unavailable). This does not include intentionally removed information, but might include, e.g., redacted text.**

In our current dataset assembly and preprocessing stages, we have striven to maintain as complete a dataset as possible for each task. However, there may be instances where some information is missing, and this is typically due to the following reasons:

- **Unavailability:** Some information might not be available at the time of data collection. For example, in financial reports or news articles, certain details about companies or their performance might not have been disclosed or might be proprietary.

- **Irrelevance:** In some cases, specific details are intentionally not included in the dataset because they do not contribute to the task at hand. An example of this is the removal of 'MISCELLANEOUS' entities in the Named Entity Recognition task, which do not contribute to the main goal of identifying persons, organizations, or locations.

- **Privacy Protection:** In the case of the datasets used for stock movement prediction, tweets related to stocks are cleaned to ensure they do not contain personal identification information or offensive content. This is done in adherence to privacy laws and regulations, and to maintain the ethical standards of the research.

- **Token Limitation:** For the ConvFinQA, FinQA, and the stock price movement prediction tasks, the instance length is restricted. Any instances exceeding a length of 2048 tokens are truncated. This is due to the limitations of transformer-based models like BERT and GPT, which can handle a maximum sequence length of 2048 tokens. This truncation may lead to the removal of some information in longer instances. However, the truncation is performed in such a manner as to preserve the most relevant information for the task at hand. Specifically, we try to maintain the context necessary for the task (e.g., the relevant financial news or question context) within the 2048 token limit.

## B.6 Are relationships between individual instances made explicit (e.g., users' movie ratings, social network links)? If so, please describe how these relationships are made explicit.

In our dataset for the domain-specific language model, FinMA, individual instances are primarily treated as standalone entries. The main focus is on the tasks to be performed on these instances, such as sentiment analysis, named entity recognition, question answering, etc. Therefore, relationships between individual instances are generally not explicitly defined. However, some inherent relationships may exist:

- **Temporal Relationships:** In the stock movement prediction task, each instance may be linked to others through their temporal ordering. For instance, stock prices and the related tweets are chronologically ordered, which introduces an implicit relationship between the instances based on their timestamp.

- **Contextual Relationships:** In the question answering tasks, including ConvFinQA and FinQA, a series of questions and their respective answers can be linked together as they are part of a conversation or discussion thread.

However, it is important to note that these relationships are intrinsic to the data and are not explicitly labelled or annotated in the dataset. The model must learn to understand and leverage these relationships from the data itself during training.

## B.7 Does the dataset contain all possible instances or is it a sample (not necessarily random) of instances from a larger set? If the dataset is a sample, then what is the larger set? Is the sample representative of the larger set (e.g., geographic coverage)? If so, please describe how this representativeness was validated/verified. If it is not representative of the larger set, please describe why not (e.g., to cover a more diverse range of instances, because instances were withheld or unavailable).

The dataset employed for FinMA is a sample drawn from a larger set of financial data sources. These sources include financial reports, news articles, stock prices, tweets, and other financial and economic texts. It is important to note that the dataset does not cover all possible instances from these sources due to the sheer volume and continuously evolving nature of such data. The selection of samples was made to enable diverse and comprehensive coverage of typical tasks in the financial domain.

- **Representativeness:** The dataset strives to be representative of the larger set of financial data sources in terms of the variety of tasks it covers (sentiment analysis, entity recognition,

question answering, stock movement prediction, etc.), modalities (text, time-series data), and types of financial text (news articles, reports, regulatory filings, etc.).

- **Validation:** The representativeness of the dataset cannot be fully validated due to the dynamic and vast nature of the larger set. However, it was designed to include diverse and important tasks in financial analysis, and its effectiveness is evaluated based on the performance of the model trained on it.

- **Limitations:** Some instances may have been excluded due to the unavailability of data, restrictions in data sharing agreements, or practical considerations such as token limitations in the model. For instance, very lengthy reports or conversations may have been truncated or excluded. Similarly, some types of financial data (e.g., confidential company reports, private communications) are inherently unavailable due to privacy and confidentiality reasons.

Despite these limitations, the dataset is intended to provide a wide coverage of tasks, modalities, and text types, and to capture the complexity and diversity of financial data analysis tasks.

## B.8 Are there recommended data splits (e.g., training, development/validation, testing)? If so, please provide a description of these splits, explaining the rationale behind them.

Yes, recommended data splits are provided for each of the tasks in the dataset, as described below:

- **Financial Sentiment Analysis:** For the FPB dataset, the data is randomly split into 3,100 instances for training, 775 instances for validation, and 970 instances for testing. Similarly, for the FiQA-SA dataset, the data is split into 750 instances for training, 188 instances for validation, and 235 instances for testing.

- **News Headline Classification:** The dataset is randomly divided into 7,988 headlines for training, 1,141 headlines for validation, and 2,283 headlines for testing.

- **Named Entity Recognition:** The NER dataset is randomly split into 408 sentences for training, 103 sentences for validation, and 98 sentences for testing.

- **Question Answering:** For the FinQA dataset, the dataset is randomly split into 6,251 instances for training, 883 for validation, 1,147 for testing. As for the ConvFinQA dataset, the training dataset is split into 2,429 samples for training and 608 samples for validation, while 421 samples from the dev dataset are adopted for testing.

- **Stock Movement Prediction:** These datasets are chronologically split into training, validation, and testing datasets following the same settings as previous methods.

The rationale behind these splits is to ensure that the model has sufficient data for training while still having a separate, unused portion of data for validation and testing. This allows us to assess the model's performance and generalizability to unseen data. The random splits for tasks such as sentiment analysis, headline classification, and NER ensure a good mix of data in each subset. For stock movement prediction, a chronological split is used instead, as the temporal sequence is crucial in this task.

## B.9 Are there any errors, sources of noise, or redundancies in the dataset? If so, please provide a description.

Given the diverse sources of data and the complex nature of financial language, there may be inherent errors, noise, or redundancies in the dataset, which include:

- **Sentiment Ambiguity:** In the Financial Sentiment Analysis task, the process of converting sentiment scores into three categories (negative, neutral, positive) may introduce noise, as sentiments expressed in financial texts can often be subjective and ambiguous.

- **Named Entity Recognition Noise:** The process of discarding miscellaneous entities in the Named Entity Recognition task might inadvertently remove useful information. Also, the presence of sentences without any entities might introduce additional noise.

- **Question Answering Complexity:** In the Question Answering task, the complexity of financial questions and their respective answers may introduce potential errors during the annotation process, given that multiple correct or partially correct answers might exist for a given question.
- **Stock Movement Prediction:** In Stock Movement Prediction, the price features and tweet data might contain redundancies, as they include multiple overlapping features such as opening, highest, lowest, closing, and adjusted closing prices. Moreover, the prediction of stock movements is influenced by a variety of factors, not all of which might be captured in the features used, leading to potential errors.
- **Token Limit:** For tasks such as ConvFinQA, FinQA, and Stock Movement Prediction, instances are truncated to 2048 tokens, which might result in loss of information and subsequently introduce errors in these tasks.

However, despite these potential issues, the datasets should still serve as valuable resources for developing and benchmarking models for financial tasks, provided these limitations are taken into account during the model development and evaluation process.

**B.10 Is the dataset self-contained, or does it link to or otherwise rely on external resources (e.g., websites, tweets, other datasets)? If it links to or relies on external resources, a) are there guarantees that they will exist, and remain constant, over time; b) are there official archival versions of the complete dataset (i.e., including the external resources as they existed at the time the dataset was created); c) are there any restrictions (e.g., licenses, fees) associated with any of the external resources that might apply to a future user? Please provide descriptions of all external resources and any restrictions associated with them, as well as links or other access points, as appropriate.**

While the dataset does originally stem from external resources such as financial news articles, tweets, and stock market data, the final version used for the tasks has been preprocessed, labelled, and stored independently, making it a self-contained dataset. This means that although the raw data was initially collected from external resources, the instances in their current format in the dataset are not directly linked to these sources, thus mitigating the issues of persistence, archival consistency, and restrictions.

The dataset is hosted on a GitHub repository, allowing for easy and open access. The nature of GitHub as a version-control platform also ensures that the dataset's state is preserved as it is at the time of its release. Users can clone or fork the repository for their use, adhering to the terms and conditions stipulated in the repository.

Thus, despite the original data being sourced externally, the final dataset is self-contained, accessible, and comes with the assurance of persistence and minimal usage restrictions.

Any other comments?

The dataset for each task exhibits a good balance between comprehensiveness and specificity. Here are some additional comments on the data composition:

- **Diversity:** The dataset exhibits a high level of diversity, especially in tasks related to financial sentiment analysis and news headline classification. The collected text data come from various sources, thus ensuring a broad coverage of financial terms and expressions.
- **Representation:** The dataset is representative of the challenges faced in the domain of financial NLP tasks. The instances are carefully chosen to ensure they simulate real-world challenges, such as sentiment analysis on financial posts, entity recognition in financial texts, question answering, and stock movement prediction.
- **Comprehensiveness:** The dataset is comprehensive in terms of the types of tasks it covers. From text classification tasks such as sentiment analysis and news headline classification to more complex tasks like question answering and stock movement prediction, the dataset provides a broad spectrum of financial NLP problems.
- **Size:** The dataset is adequately large for training robust models. For each task, a substantial number of instances are available, supporting model training, validation, and testing.

These factors contribute to the utility and value of the dataset in the context of financial LLMs.

## C    Collection Process

### C.1    What mechanisms or procedures were used to collect the data (e.g., hardware apparatus or sensor, manual human curation, software program, software API)? How were these mechanisms or procedures validated?

The dataset has been assembled from publicly available datasets and resources, supplemented by manual creation of instruction templates. More specifically:

- **Publicly Available Datasets:** Existing datasets have been utilized in our tasks, including financial sentiment analysis, news headline classification, named entity recognition, and question answering. These datasets are widely acknowledged and utilized in the research community, validating their reliability and quality. The raw datasets are provided along with our work for reference.
- **Manual Creation of Instruction Templates:** For tasks that rely on conversational models, we have manually created instruction templates. These templates guide the model's responses and ensure that the output is consistent with the task's requirements.

The full details, including the source of the raw datasets, are provided in the documentation accompanying our GitHub repository. The use of publicly available datasets, coupled with manual curation of instructions, helps ensure the data's reliability and robustness for our tasks.

### C.2    How was the data associated with each instance acquired? Was the data directly observable (e.g., raw text, movie ratings), reported by subjects (e.g., survey responses), or indirectly inferred/derived from other data (e.g., part-of-speech tags, model-based guesses for age or language)? If data was reported by subjects or indirectly inferred/derived from other data, was the data validated/verified? If so, please describe how.

We constructed the multi-task and multi-modal instruction data by collecting publicly available training data from a range of diverse tasks. This data was directly observable and derived from multiple open released financial datasets. The data we used included both textual and time-series data modalities, which allowed our model to handle tasks such as sentiment analysis, news headline classification, named entity recognition, question answering, and stock movement prediction. For each task, we wrote specific instructions that were combined with the data samples to create our large-scale instruction tuning data.

The instructions for each task were carefully designed by domain experts to ensure that they accurately reflect the nuances and requirements of the different tasks. This approach allowed us to tailor our large language model, FinMA, to perform a diverse range of financial tasks.

### C.3    If the dataset is a sample from a larger set, what was the sampling strategy (e.g., deterministic, probabilistic with specific sampling probabilities)?

Our dataset is not a sample from a larger dataset but an assembly of publicly available multi-task and multi-modal data derived from multiple open-released financial datasets. We didn't use a specific sampling strategy since we weren't sampling from a larger set. Instead, we collected the complete datasets that were available and relevant to our study

### C.4    Who was involved in the data collection process (e.g., students, crowdworkers, contractors) and how were they compensated (e.g., how much were crowdworkers paid)?

For the PIXIU project, our data collection process involved both domain experts and students. The domain experts, who are professionals in the financial field such as fund managers, were invited to design the task-specific instructions for each dataset. On the other hand, students were responsible for the collection of publicly available datasets.

This project was conducted as collaborative research, so those who contributed to the data collection and instruction design were not compensated in a traditional paid manner. Instead, they contributed their expertise and time to the project as part of their research activities or professional engagement.

**C.5 Over what timeframe was the data collected? Does this timeframe match the creation timeframe of the data associated with the instances (e.g., recent crawl of old news articles)? If not, please describe the timeframe in which the data associated with the instances was created.**

- FPB, while it does not provide a specific timeframe for data collection, was published in 2014.
- FIQASA, like FPB, does not specify a timeframe for data collection, but it was published in 2018.
- The Headlines dataset contains human-annotated news headlines that were collected over a span of 19 years, from 2000 to 2019. It was published in 2021.
- NER, again, does not indicate a specific timeframe for data collection, but it was published in 2015.
- The FinQA dataset spans two decades, collecting data from 1999 to 2019. It was published in 2021.
- ConvFinQA, like FinQA, covers a period from 1999 to 2019 and was published in 2022.
- The BigData22 dataset has a more confined range of data collection, from July 5th, 2019 to June 30th, 2020.
- The ACL18 dataset spans approximately two years, from January 2nd, 2014 to December 30th, 2015.
- The CIKM18 dataset covers data collected within a single year, from January 3rd, 2017 to December 28th, 2017.

In summary, the data associated with the instances in the PIXIU dataset spans a wide range of timeframes, from specific periods within a year to a stretch of 20 years. Some of these match the publication years of their respective sources, like the Headlines dataset, FinQA, and ConvFinQA, while others, like FPB, FIQASA, and NER, do not specify a timeframe for data collection. Therefore, the creation timeframe of the data associated with the instances varies, and in some cases, it may be assumed to be close to their respective publication years.

## D  Data Preprocessing

**D.1 Was any preprocessing/cleaning/labeling of the data done (e.g., discretization or bucketing, tokenization, part-of-speech tagging, SIFT feature extraction, removal of instances, processing of missing values)? If so, please provide a description. If not, you may skip the remainder of the questions in this section.**

To convert raw data into a structured instruction dataset, we defined a pipeline with the following steps:

1. **Define Instruction Templates:** We began by defining clear and succinct instruction templates for each task.
2. **Extract Relevant Information:** The next step involved extracting the requisite pieces of information from the raw data. These would be used to fill in the instruction templates.

3. **Match to Template:** Once the relevant information was extracted, it was matched to the instruction template.

4. **Verify and Clean Instructions:** Following the creation of the instructions, we verified that they were both coherent and accurate representations of the task.

5. **Standardize Instructions:** Finally, we standardized the instructions in terms of language, structure, and style. This ensured consistency across the dataset, which subsequently enabled the model to understand and learn the tasks more efficiently.

This pipeline was utilized to process each task within the PIXIU model. For example, consider a task involving sentiment analysis. The raw data consisted of a series of tweets along with their associated sentiment labels. The instruction template for this task could be: "Classify the sentiment of the following tweet as 'positive', 'negative', or 'neutral'". The preprocessing steps involved extracting the tweet text, fitting it into the template, and verifying that the resulting instruction accurately represented the task.

## D.2 Was the "raw" data saved in addition to the preprocessed/cleaned/labeled data (e.g., to support unanticipated future uses)? If so, please provide a link or other access point to the "raw" data.

The raw data can be accessed through the original papers:

1. Pekka Malo, Ankur Sinha, Pekka Korhonen, Jyrki Wallenius, and Pyry Takala. 2014. Good debt or bad debt: Detecting semantic orientations in economic texts. Journal of the Association for Information Science and Technology 65, 4 (2014), 782–796.

2. Macedo Maia, Siegfried Handschuh, André Freitas, Brian Davis, Ross McDermott, Manel Zarrouk, and Alexandra Balahur. 2018. Www'18 open challenge: financial opinion mining and question answering. In Companion proceedings of the the web conference 2018. 1941–1942

3. Ankur Sinha and Tanmay Khandait. 2021. Impact of news on the commodity market: Dataset and results. In Advances in Information and Communication: Proceedings of the 2021 Future of Information and Communication Conference (FICC), Volume 2. Springer, 589–601.

4. Julio Cesar Salinas Alvarado, Karin Verspoor, and Timothy Baldwin. 2015. Domain adaption of named entity recognition to support credit risk assessment. In Proceedings of the Australasian Language Technology Association Workshop 2015. 84–90.

5. Zhiyu Chen, Wenhu Chen, Charese Smiley, Sameena Shah, Iana Borova, Dylan Langdon, Reema Moussa, Matt Beane, Ting-Hao Huang, Bryan R Routledge, et al . 2021. FinQA: A Dataset of Numerical Reasoning over Financial Data. In Proceedings of the 2021 Conference on Empirical Methods in Natural Language Processing. 3697–3711.

6. Zhiyu Chen, Shiyang Li, Charese Smiley, Zhiqiang Ma, Sameena Shah, and William Yang Wang. 2022. Convfinqa: Exploring the chain of numerical reasoning in conversational finance question answering. arXiv preprint arXiv:2210.03849 (2022).

7. Yejun Soun, Jaemin Yoo, Minyong Cho, Jihyeong Jeon, and U Kang. 2022. Accurate Stock Movement Prediction with Self-supervised Learning from Sparse Noisy Tweets. In 2022 IEEE International Conference on Big Data (Big Data). IEEE, 1691–1700.

8. Yumo Xu and Shay B Cohen. 2018. Stock movement prediction from tweets and historical prices. In Proceedings of the 56th Annual Meeting of the Association for Computational Linguistics (Volume 1: Long Papers). 1970–1979.

9. Huizhe Wu, Wei Zhang, Weiwei Shen, and Jun Wang. 2018. Hybrid deep sequential modeling for social text-driven stock prediction. In Proceedings of the 27th ACM international conference on information and knowledge management. 1627–1630.

### D.3 Is the software used to preprocess/clean/label the instances available? If so, please provide a link or other access point.

Yes, the software used for preprocessing, cleaning, and labeling the instances is publicly available. It can be accessed via our GitHub repository at https://github.com/chancefocus/PIXIU.

### D.4 Does this dataset collection/processing procedure achieve the motivation for creating the dataset stated in the first section of this datasheet? If not, what are the limitations?

Yes, the dataset collection and processing procedure does fulfill the initial motivation. We have created a comprehensive set of resources specifically tailored for the financial sector. However, as with any novel project, there are limitations. The financial domain is vast and complex, and while PIXIU covers a broad range of topics, it may not encompass all possible financial tasks or scenarios. We hope that future iterations of PIXIU can further expand and diversify the instruction dataset and evaluation benchmarks.

### D.5 Any other comments

No.

## E Dataset Distribution

### E.1 How will the dataset be distributed? (e.g., tarball on website, API, GitHub; does the data have a DOI and is it archived redundantly?)

The dataset, along with the necessary code, will be distributed through our GitHub repository (https://github.com/chancefocus/PIXIU).

### E.2 When will the dataset be released/first distributed? What license (if any) is it distributed under?

The dataset was first released on June 11th, 2023. It is distributed under the MIT License, which permits use, copy, modify, merge, publish, distribute, sublicense, and/or sell copies of the Software with few restrictions, provided that the above copyright notice and this permission notice are included in all copies or substantial portions of the Software.

### E.3 Are there any copyrights on the data?

The transformed and aggregated dataset, as well as the code and models we provide, are released under the MIT License.

### E.4 Are there any fees or access/export restrictions?

There are no fees associated with the PIXIU, and it can be freely accessed and exported for research purposes.

### E.5 Any other comments?

We encourage researchers to use this dataset and related resources to advance the field of financial AI. However, proper attribution should be given when using these resources in line with standard academic practices.

## F  Dataset Maintenance

### F.1  Who is supporting/hosting/maintaining the dataset?

The dataset is maintained and supported by a team of researchers from various institutions. The team includes Qianqian Xie, Weiguang Han, and Min Peng from the School of Computer Science, Wuhan University, Xiao Zhang from Sun Yat-Sen University, Yanzhao Lai from the School of Economics and Management, Southwest Jiaotong University, Alejandro Lopez-Lira from the University of Florida, and Jimin Huang from ChanceFocus AMC.

### F.2  Will the dataset be updated? If so, how often and by whom?

The updates will be managed by Weiguang Han from the School of Computer Science, Wuhan University. The frequency of updates will be determined by the availability of new data and feedback from users.

### F.3  How will updates be communicated? (e.g., mailing list, GitHub)

Updates about the dataset will be communicated through GitHub, where the dataset is hosted. Users are encouraged to keep track of the GitHub repository for any updates.

### F.4  If the dataset becomes obsolete how will this be communicated?

If the dataset becomes obsolete, this will be communicated through the same GitHub repository where the dataset is hosted.

### F.5  Is there a repository to link to any/all papers/systems that use this dataset?

Currently, there is no specific repository for linking to papers or systems that use the dataset. However, users are encouraged to cite the PIXIU paper in their work.

### F.6  If others want to extend/augment/build on this dataset, is there a mechanism for them to do so? If so, is there a process for tracking/assessing the quality of those contributions. What is the process for communicating/distributing these contributions to users?

The dataset is open source and users are welcome to extend, augment, or build on it. Users can submit pull requests to the GitHub repository where the dataset is hosted. The quality of contributions will be assessed by the maintenance team before being integrated into the dataset. These contributions, once approved, will be made available to other users through the GitHub repository.

## G  Legal and Ethical Considerations

### G.1  Were any ethical review processes conducted (e.g., by an institutional review board)? If so, please provide a description of these review processes, including the outcomes, as well as a link or other access point to any supporting documentation.

No specific ethical review processes were conducted for the development of PIXIU as it leverages previously published and publicly accessible datasets. These original datasets underwent their own ethical review processes.

**G.2 Does the dataset contain data that might be considered confidential (e.g., data that is protected by legal privilege or by doctorpatient confidentiality, data that includes the content of individuals non-public communications)? If so, please provide a description.**

The datasets used in PIXIU do not contain any data that might be considered confidential. Any potential personal information in the original datasets has been removed by the original authors before they were made publicly accessible.

**G.3 Does the dataset contain data that, if viewed directly, might be offensive, insulting, threatening, or might otherwise cause anxiety? If so, please describe why**

The datasets used in PIXIU do not contain data that, if viewed directly, might be offensive, insulting, threatening, or might otherwise cause anxiety. The original authors of the datasets ensured this during their preprocessing and cleaning stages.

**G.4 Does the dataset relate to people? If not, you may skip the remaining questions in this section.**

While some of the datasets may originally have pertained to people, for instance, those containing tweets, any direct identifiers have been removed from the original datasets. Therefore, PIXIU does not directly relate to identifiable individuals.

**G.5 Does the dataset identify any subpopulations (e.g., by age, gender)? If so, please describe how these subpopulations are identified and provide a description of their respective distributions within the dataset.**

No.

**G.6 Is it possible to identify individuals (i.e., one or more natural persons), either directly or indirectly (i.e., in combination with other data) from the dataset? If so, please describe how.**

It is not possible to identify individuals either directly or indirectly from the datasets used in PIXIU. All potentially identifiable information was removed from the original datasets before they were made publicly accessible.

**G.7 Does the dataset contain data that might be considered sensitive in any way (e.g., data that reveals racial or ethnic origins, sexual orientations, religious beliefs, political opinions or union memberships, or locations; financial or health data; biometric or genetic data; forms of government identification, such as social security numbers; criminal history)? If so, please provide a description.**

The datasets used in PIXIU do not contain data that might be considered sensitive in any way. All potentially sensitive information was removed from the original datasets before they were made publicly accessible.

**G.8 Did you collect the data from the individuals in question directly, or obtain it via third parties or other sources (e.g., websites)?**

The data used in PIXIU was not directly collected from individuals. Instead, it was obtained from previously published datasets that were collected by other researchers and made publicly available.

**G.9 Were the individuals in question notified about the data collection? If so, please describe (or show with screenshots or other information) how notice was provided, and provide a link or other access point to, or otherwise reproduce, the exact language of the notification itself.**

As the data used in PIXIU were obtained from previously published datasets, the original data collectors were responsible for notifying individuals about the data collection.

**G.10 Did the individuals in question consent to the collection and use of their data? If so, please describe (or show with screenshots or other information) how consent was requested and provided, and provide a link or other access point to, or otherwise reproduce, the exact language to which the individuals consented.**

As the data used in PIXIU were obtained from previously published datasets, the original data collectors were responsible for obtaining consent from individuals for the collection and use of their data.

**G.11 If consent was obtained, were the consenting individuals provided with a mechanism to revoke their consent in the future or for certain uses? If so, please provide a description, as well as a link or other access point to the mechanism (if appropriate).**

The issue of revoking consent does not directly apply to PIXIU as the data used was obtained from previously published datasets. The original data collectors were responsible for providing mechanisms for revoking consent.

**G.12 Has an analysis of the potential impact of the dataset and its use on data subjects (e.g., a data protection impact analysis)been conducted? If so, please provide a description of this analysis, including the outcomes, as well as a link or other access point to any supporting documentation.**

Given that the data used in PIXIU is de-identified and does not contain sensitive information, it is not anticipated that there will be direct impacts on data subjects. However, as with any research involving human-related data, there is always a responsibility to use the data ethically and with respect to potential implications.

**G.13 Any other comments?**

No.

## H  Responsibility Statement

The authors of PIXIU bear all responsibility in case of any violation of rights or any other legal issues that arise from the use of this dataset. The authors have taken all possible measures to ensure the respect of privacy and ethical guidelines in the construction of this dataset.

The datasets included in PIXIU are distributed under the MIT license. By using these datasets, users agree to comply with the terms of this license.

This paper, including any associated source codes, datasets, and appendices ("Material"), is intended solely for academic and educational purposes. The Material does not constitute financial, legal, or investment advice and is not intended to be a basis for any decision-making.

While the authors have taken reasonable measures to ensure the accuracy of the Material, no warranty, express or implied, is made as to its completeness, reliability, or suitability for any specific purpose. The authors and their affiliated organizations shall not be liable for any losses, damages, or consequences, whether direct or indirect, arising from the use or reliance on the Material. It is

the responsibility of the user to consult with professionals for any financial, legal, or investment decisions.

By referencing or utilizing this Material, the reader agrees to indemnify, defend, and hold harmless the authors and any affiliated organizations or persons from any and all claims or damages arising from such use.

# I   Reproducibility

Ensuring the reproducibility of research results is of utmost importance in promoting transparency and enabling further scientific advancements. In the context of the benchmarks presented in this datasheet, we have taken several measures to facilitate the reproducibility of our reported results.

To begin with, we have made all the necessary code, data, and instructions available in the PIXIU GitHub repository (`https://github.com/chancefocus/PIXIU`). This repository serves as a centralized hub where researchers can access the resources required to reproduce the benchmarks. The repository is organized and well-documented, providing a clear and structured framework for replication.

In order to enhance reproducibility, we have adhered to the ML reproducibility checklist, a framework that promotes best practices for ensuring the replicability of machine learning experiments. By following this checklist, we have prioritized the inclusion of all essential components, such as code, datasets, and evaluation procedures, making it easier for researchers to reproduce our reported results.

Furthermore, we have provided detailed instructions within the repository, outlining the steps needed to replicate the benchmarks. These instructions serve as a guide for researchers, ensuring that they have the necessary information and resources at their disposal to validate and verify our findings.

We encourage researchers to leverage the resources available in the PIXIU GitHub repository to replicate our benchmarks and explore further extensions and improvements. By promoting a culture of reproducibility, we aim to foster collaboration and drive advancements in the field of financial artificial intelligence.