# OpenReview forum: "PIXIU: A Comprehensive Benchmark, Instruction Dataset and Large Language Model for Finance"
_NeurIPS.cc/2023/Track/Datasets_and_Benchmarks — NeurIPS 2023 Datasets and Benchmarks Poster_

### Official Review · Reviewer_7TUR · 2023-07-17
**A collection of financial NLP benchmark datasets and instruction tuning with LLAMA-7B/13B models**

**Rating:** 5
**Confidence:** 4
**Correctness:** The evaluation is made in a technical…
**Clarity:** This is, in general, clearly written …

**Strengths:**

- This paper comprehensively evaluates the existing general LLMs, like ChatGPT and GPT-4, and the existing finance domain LLMs, like BloombergGPT, including 9 datasets across 5 tasks.

- This paper released a set of instruction-tuning datasets curated based on public financial NLP datasets, which offers a way for future research to build instruction-tuned LLMs for finance.

- This paper open-sourced the instruction-tuned LLM models based on LLAMA-7B and 13B.

**Additional Feedback:**

Table 5: what are those "-" markers indicating for the results?

**Documentation:**

The datasets listed in the paper are available at the GitHub repo link shown in the paper.

**Ethics:**

No ethics concern is raised for this paper.

**Limitations:**

Please refer to the section "Opportunities For Improvement."

**Opportunities For Improvement:**

- The selected baselines are all generative/decoder-only LLMs like GPT compared with the proposed FinMA. Nonetheless, it is suggested to involve those bidirectional LLMs, like FinBERT [1] and FLANG [2], especially for the tasks that bidirectional LLMs are expected to perform well, like the binary classification tasks and named entity recognition tasks. According to [2], BERT-based methods can achieve very high scores for financial sentiment classification tasks (about 0.9 ACC), headline classification (about  0.98 F-1), NER (about  0.8 F-1), etc. On the other side, it is observed that the selected generative LLMs, including the proposed FinMA, achieved much lower absolute scores on these tasks, though the datasets are not identical to the ones used in [2].

- On the selected stock movement prediction benchmarks (BigData22, ACL18, and CIKM18), none of the selected methods get results significantly better than random guess results (about  0.5 ACC). It would be beneficial to test other prediction models for these tasks, e.g., XGBoost, to offer a sense of the difficulty of these benchmarks.

- There is no sufficient argument that supports the results: BloombergGPT, pre-trained on large-scale financial texts with 50B model size, is much worse than the FinMA-7B/13B models, while the latter only undertakes fine-tuning on around 100K samples.

[1] Yi Yang, Mark Christopher Siy Uy, and Allen Huang. 2020. Finbert: A pre-trained language model for financial communications. arXiv preprint arXiv:2006.08097 (2020).

[2] Raj Sanjay Shah, Kunal Chawla, Dheeraj Eidnani, Agam Shah, Wendi Du, Sudheer Chava, Natraj Raman, Charese Smiley, Jiaao Chen, and Diyi Yang. 2022. WHEN FLUE MEETS FLANG: Benchmarks and Large Pre-trained Language Model for Financial Domain. arXiv preprint arXiv:2211.00083 (2022).

**Relation To Prior Work:**

I believe this work provides the necessary arguments that differentiate this work from the prior arts.

**Summary And Contributions:**

This paper open-sourced a collection of

- instruction tuning datasets for large language models (LLMs) specifically for financial NLP and prediction tasks

- a suite of fine-tuned financial LLMs based on LLAMA-7B/13B, called (FinMA)

- the benchmarking results on 9 datasets spreading 5 tasks: sentiment analysis, news headline classification, named entity recognition, question answering, and stock movement prediction.

As a result, the authors identified that FinMA wins over other LLMs like BloombergGPT, ChatGPT, GPT-4 on 4 out of 5 tasks, excluding the question-answering task.

---

> ### Author Response · Authors · 2023-08-22
>
> Dear Reviewer 7TUR,
>
> We would like to extend our sincere gratitude for the comprehensive and thoughtful review of our manuscript. Your comments are valuable in helping us improve the quality of our work.
>
> **Include bidirectional LLMs baselines**
> To address the concern, we have included the results and analysis of FinBERT and FLANG in the Appendix due to the space limit. While FinBERT and FLANG have higher performance in specific tasks when fine-tuned, their performance boost may be partly due to overfitting, as they require supervised fine-tuning for each specific task. Additionally, these models are limited to tasks pre-defined in their training data, which hinders their generalization ability. In contrast, generative LLMs like our proposed FinMA have a zero-shot capability, allowing them to handle tasks not seen during training. They also accept natural language prompts, allowing instruction-tuning, which helps to unify various tasks during supervised fine-tuning and is more user-friendly.
>
> **Demonstrating the difficulty of the stock movement prediction task**
> To address this concern, we have included the results of non-LLM models such as XGBoost and Logistic regression (LR) et al, in the Appendix, Table 2, due to space constraints. For the task, traditional models have long been employed, and also face significant challenges in achieving consistently high levels of accuracy. This underscores the inherent difficulty of the task at hand. Different from other tasks such as Financial QA, the task is not just about numerical reasoning; it's a complex task influenced by various qualitative factors. This complexity is evidenced by ChatGPT's good performance in numeric reasoning but poor results in stock prediction.
>
> **FinMA vs BloombergGPT**
> From table 4, we can figure out FinMA outperforms, BloombergGPT on traditional financial nlp tasks. Since BloombergGPT and its test data are not open-released. It is difficult to have a comprehensive comparison with it. This is one of the main motivations of our paper, we consider the value of our work as being the pioneer in releasing open-source instruction-tuning data, setting up a comprehensive evaluation benchmark, introducing a specialized financial LLM, and highlighting both the advantages and limitations of LLMs on the financial domain. We sincerely hope these contributions will advance transparency and development in the field of financial AI.
>
> Once again, we'd like to extend our sincere thanks to your insightful comments and suggestions. Your feedback has been invaluable in refining our work and addressing key issues.
>
> Best regards,
>
> PIXIU Authors

---

> > ### Comment · Reviewer_7TUR · 2023-08-29
> >
> > I appreciate the authors' efforts in addressing the concerns. However, I believe the concern about the performance gap between the decoder-only LLM and other models on the selected tasks remains. I adjust the score to reflect the improvement of the manuscript.

---

> > > ### Author Response · Authors · 2023-08-29
> > >
> > > Dear Reviewer 7TUR,
> > >
> > > We are sincerely grateful for your recognition of the improvements we have made to our manuscript and your positive adjustment of its score. Your insightful comments have been vital in enhancing the quality of our research.
> > >
> > > For your particular concern about the performance gap between decoder-only Large Language Models (LLMs) and Bidirectional Pretrained Language Models (PLMs), we view this gap as a pivotal finding of our study. Our results and analysis indicate that PLMs like FinBERT and FLANG, often pre-trained on domain-specific datasets, outperform in specialized tasks but exhibit limitations on adaptability across multiple tasks and zero-shot learning. In contrast, LLMs may not achieve the same level of performance but display superior adaptability in multi-task scenarios and have the advantage of learning directly from prompts.
> > >
> > > Our observations align closely with those of Shah et al., creators of the FLANG model [1]. Their research, although limited to four datasets, shows that zero-shot LLMs like ChatGPT yield commendable performance across financial tasks, even without labeled data. However, they note that this performance drops when compared to specialized PLMs, particularly for datasets that are not publicly accessible. Shah et al. also highlight that the use of generative LLMs necessitates a significantly greater time commitment for manual labeling, in contrast to fine-tuned PLMs. Our analysis also aligns with the study by Jingwei Ni et al. [2], which shows that even when PLMs are adapted for multi-task learning through task-specific headers, they still struggle to adapt to unforeseen tasks.
> > >
> > > Expanding upon these studies, we take an innovative approach by being the first to provide a comprehensive empirical framework for the direct comparison of financial PLMs and LLMs. Our work pioneers the development of both an evaluation framework and an instructional dataset and extends to fine-tuning LLMs in a multi-task learning environment. One research aim of this study is to explore the potential and limitations of LLMs, and our results show the great potential of LLMs in scenarios involving multiple tasks and also emphasize the need for domain-specific pretraining to improve their applicability in specialized domains like finance.
> > >
> > > Owing to space limitations, we have updated the above discussion to the appendix,  and are currently conducting experiments to explore the performance of BERT-based PLMs on other tasks within our benchmark, to provide a more exhaustive comparison. The results of these experiments will be reported once they are yielded.
> > >
> > > We genuinely value your thoughtful feedback and are confident that this amended version more thoroughly addresses your concerns.
> > >
> > > With sincere regards,
> > >
> > > PIXIU Authors
> > >
> > > References:
> > >
> > > [1] Shah, Agam, and Sudheer Chava. “Zero is Not Hero Yet: Benchmarking Zero-Shot Performance of LLMs for Financial Tasks.” ArXiv abs/2305.16633 (2023): n. pag.
> > >
> > > [2] Jingwei Ni, Zhijing Jin, Qian Wang, Mrinmaya Sachan, and Markus Leippold. 2023. "When Does Aggregating Multiple Skills with Multi-Task Learning Work? A Case Study in Financial NLP." In Proceedings of the 61st Annual Meeting of the Association for Computational Linguistics (Volume 1: Long Papers), pages 7465–7488, Toronto, Canada. Association for Computational Linguistics.

---

### Official Review · Reviewer_TmzR · 2023-07-20
**Good work exploring potential ability of LLMs on financial tasks**

**Rating:** 7
**Confidence:** 3
**Correctness:** The claims about dataset and benchmar…
**Clarity:** This paper is well written and organi…

**Strengths:**

1. This work presents a comprehensive framework including the first financial LLM based on fine-tuning LLaMA with instruction data, the first instruction data with 136K data samples to support the fine-tuning, and an evaluation benchmark with 5 tasks and 9 datasets.
2. Extensive evaluation results demonstrate the effectiveness of FinMA across various financial tasks, showing the potential of domain -specific instruction tuning of large language models in the financial domain. This open-source contribution is expected to facilitate further research and innovation in financial LLM in the field of finance.

**Additional Feedback:**

No.

**Documentation:**

The authors have included Motivation for Datasheet Creation, Datasheet Composition, Collection Process, Data Preprocessing Methods, Dataset Distribution, Dataset Maintenance Statement, Legal and Ethical Considerations, Responsibility Statement and Reproducibility Supporting Document in the supplementary material.

**Ethics:**

No.

**Limitations:**

The authors have adequately addressed the limitations and negative societal impact of this work.

**Opportunities For Improvement:**

1. As a work about benchmark, the possible improvement is to design and evaluate more wide range of tasks, especially more difficult ones like financial prediction tasks. The authors have mentioned that their FinMA model is still struggling on financial prediction tasks.
2. Larger models should be evaluated on the proposed benchmark FLARE.

BTW, some typos are found in the article:
1. In line 14, you mentioned “including five financial NLP tasks and one financial prediction task. ” Should it be “four financial NLP tasks and one financial prediction task.”?
2. In line 243, you mentioned "Vicuna-13B" is a baseline model compared with FinMA, but no results of Vicuna-13B in the article, just in Table 6 it shows the EmAcc of Vicuna-7B on ConvFinQA task. Could you please check it?
3. In line 270, “FLAPE ” should be “FLARE”.

**Relation To Prior Work:**

Yes, it compares their work with prior work like BloombergGPT, finBERT, FinBERT, etc.

**Summary And Contributions:**

This work presents PIXIU, consisting of an instruction tuning dataset FIT, the first open-sourced financial large language model FinMA which is instruction-tuned from LLaMA model using FIT dataset, and an evaluation benchmark FLARE covering financial NLP tasks and prediction tasks. To be specific,
1) It introduces FIT, the first multitask and multi-modal instruction tuning data in the financial domain, covering 5 tasks and 9 datasets with 136,609 samples.  The 5 tasks include sentiment analysis, news headline classification, named entity recognition, question answering, and stock movement prediction.
2) It introduces FLARE, the first evaluation benchmark with both financial natural language understanding and prediction tasks.
3) It introduces FinMA, the first open-sourced instruction-following financial large language model, which achieves SOTA on 3 financial NLP tasks and 1 financial prediction task.
4) It compares FinMA and existing LLMs, BloombergGPT, GPT-4, ChatGPT, BLOOM, GPT-NeoX, OPT-66B, and vicuna-7B, on FLARE. The results demonstrate the superiority of FinMA, the key limitations of LLMs for finance, and future directions to advance LLMs for finance.

---

> ### Author Response · Authors · 2023-08-22
>
> Dear Reviewer TmzR,
>
> Thank you for your thorough review and invaluable feedback. We have incorporated them into the revised version of our paper.
>
> **Expand and evaluate more tasks**
>  In the revised version of the paper, we have expanded our benchmark to include one financial prediction task with two new datasets, and three financial NLP tasks with four additional datasets: 1) Credit Scoring, which offers a more stable evaluative framework than stock prediction by focusing on consumer credit risk. 2) Financial Text Summarization: a generative task designed to assess the model's generation ability and ability to maintain factual consistency. To address factuality and faithfulness concerns, BARTScore has been employed for our text summarization task. 3) Hawkish-Dovish Classification: a more challenging classification task that tests the model to understand nuanced financial language and its economic implications. For a more in-depth exploration of the datasets, kindly refer to Section 4. We deeply appreciate your feedback and remain committed to continually refining our evaluative approaches.
>
> **Larger models evaluations**
> In response to this valuable suggestion, we've undertaken experiments and incorporated the results for GPT-NeoX in the Appendix and finma-30B-nlp in Table 5. We fully recognize the significance of evaluating the performance of large LLMs such as LLaMA2 70B and BLOOMZ 176B. However, due to the time-intensive nature of these evaluations, which can span several months for comprehensive testing across all our tasks, they could not be concluded within this month. Nevertheless, these evaluations are actively underway, and once finalized, we commit to promptly updating the results on our GitHub repository and leaderboard. We expect these evaluations to be completed in the upcoming months.
>
> **Line 14 typo** In our revised version, we've updated the content to reflect "six financial NLP tasks and two financial prediction tasks", in light of the additional datasets we've incorporated.
>
> **Vicuna-13B typo** Our apologies for the oversight. The mention of "Vicuna-13B" in line 243 was indeed a mistake. We've corrected it to "Vicuna-7B" in the revised version.
>
> **FLARE typo** We've corrected this typo in line 270 in the revised version.
>
> Thank you once again for your valuable feedback, which helps us improve the quality and impact of our paper.
>
> Best regards,
>
> PIXIU Authors

---

> > ### Comment · Reviewer_TmzR · 2023-08-30
> > **Response to rebuttal**
> >
> > Thanks to the authors for the improvement of the manuscript.

---

> > > ### Author Response · Authors · 2023-08-30
> > >
> > > Dear Reviewer TmzR,
> > >
> > > We would like to extend our sincerest thanks for acknowledging the improvements made to the manuscript. Your insightful comments and constructive criticisms during the initial review process have been invaluable in enhancing the quality of our work.
> > >
> > > We are pleased to know that you find the revised manuscript to be improved, and we are grateful for the time and effort you have invested in reviewing our paper.
> > >
> > > Thank you once again for your constructive feedback and for recognizing the improvements in the manuscript.
> > >
> > > Sincerely,
> > >
> > > PIXIU Authors

---

### Official Review · Reviewer_v2GB · 2023-07-21
**Good topic, fair experimental design, but unreasonable task selections**

**Rating:** 5
**Confidence:** 4
**Clarity:** Not too bad.

**Strengths:**

Topic: In the era of LLM, some researchers argue that deploying the generalization ability of LLMs into downstream domains is more important than the model architecture design. This work lies on the topic of FinLLM, which is a suitable testbed.

Resource: The authors released their datasets on Github, covering 4 financial NLP tasks with 6 datasets, and 1 financial prediction task with 3 datasets. Although most of the datasets are originally contributed by others.

Evaluation: The authors are aware of the importance of evaluation towards LLM. As LLMs continue to play a vital role in both research and daily use, their evaluation becomes increasingly critical, not only at the task level but also at the society level for a better understanding of their potential risks, especially in Finance, which is a high-stake domain.





**Additional Feedback:**

None.

**Correctness:**

The dataset is constructed in a sound way. The benchmark can be extended in several ways, which can be seen in Limitations.

**Documentation:**

There is sufficient detail on data collection, but lacks the consideration of ethical and responsible use.

**Ethics:**

No.

**Limitations:**

1. As mentioned before, as a responsible AI model, the prediction of the stock market should be accompanied by risk warnings.

2. The evaluation can be carefully designed. The current evaluation design of FinLLM is too simple to be effective. The factuality and faithfulness can not be measured by simply conducting evaluations on FinNLP and stock prediction tasks.

3. Methodology: Fine-tuning LLaMA with downstream instructions is not an exciting method in the era of LLM.

**Opportunities For Improvement:**

Regarding the experiments, the stock prediction task may not be the most suitable one to evaluate the ability of the FinLLM because of the high volatility and uncertainty of the financial market. Also, as a responsible AI model, the prediction of the stock market should be accompanied by risk warnings. Besides, the evaluation of FinLLM can be improved and not limited to the topic of traditional NLP classification and stock movement prediction tasks.



**Relation To Prior Work:**

A new topic with a novel model.

**Summary And Contributions:**

This paper introduces PIXIU, a novel financial LLM based on LLAMA with instruction data. In addition to the model implementation, they propose a new benchmark that consists of five FinNLP tasks, along with a stock prediction task, to support the evaluation of financial LLMs. The most commendable credit in this paper is that the authors consider the evaluation benchmark as a crucial step towards building FinLLMs, which usually be ignored by previous work. In contrast, my major concern is that compared with the evaluation tasks used in FLUE, this work only extends one task, which is the stock movement prediction task. However, using whether the PLMs or LLMs which are trained with historical data to predict the stock movement in the future is not convincible enough to support the fair evaluation of FinLLMs.

---

> ### Author Response · Authors · 2023-08-22
>
> Dear Reviewer v2GB,
>
> Thank you for your thoughtful review of our paper and for highlighting both its strengths and areas for improvement.
>
> **the stock prediction task may not be the most suitable evaluation task, improving with new tasks, FinLLM evaluation too simplistic:**
> To address these concerns, in the revised version of the paper, we have expanded our benchmark to include one financial prediction task with two new datasets, and three financial NLP tasks with four additional datasets: 1) Credit Scoring, which offers a more stable evaluative framework than stock prediction by focusing on consumer credit risk. 2) Financial Text Summarization: a generative task designed to assess the model's generation ability and ability to maintain factual consistency. 3) Hawkish-Dovish Classification: a more challenging classification task that tests the model to understand nuanced financial language and its economic implications. To address factuality and faithfulness concerns, BARTScore has been employed for our text summarization task. However, assessing these attributes in financial LLMs remains an intricate challenge requiring collaborative efforts from academia and industry. We will continue to refine our evaluation methods and welcome further insights.
>
> **AI stock predictions need warnings:**
> We completely agree that risk warnings are crucial when applying financial LLMs to financial tasks. We've explicitly addressed the potential risks and limitations of our model in the "Limitations" section of our paper, the appendix and on our GitHub page, where we indicate that the model is currently best suited for research purposes only. We'll limit our model to include risk reminders during financial forecasts. Future updates will focus on improving factuality and reducing bias through continual financial training and RLHF for human value alignment.
>
> **Fine-tuning LLaMA unexciting:**
> Thank you for your thoughtful comment, and we appreciate the opportunity to clarify the focus of our work. While we don't consider the fine-tuning of LLaMA to be our primary contribution, we consider the value of our work is being the pioneer in releasing open-source instruction-tuning data, setting up a comprehensive evaluation benchmark, introducing a specialized financial LLM, and highlighting both the advantages and limitations of LLMs on the financial domain. We sincerely hope these contributions will advance transparency and development in the field of financial AI.
>
> We believe that addressing these concerns will significantly enhance the quality and impact of our paper. Once again, thank you for your constructive feedback and suggestions.
>
> Best regards,
>
> PIXIU authors

---

> > ### Comment · Reviewer_v2GB · 2023-08-30
> >
> > I appreciate the authors' detailed response in addressing the problems. However, I believe the concern about the scientific values of fine-tuning LLaMA on a downstream domain remains. I will keep the score to reflect the assessment.

---

> > > ### Author Response · Authors · 2023-08-30
> > >
> > > Dear Reviewer v2GB,
> > >
> > > We are grateful for your feedback with our work. Your instructions has been pivotal in enhancing the quality of our manuscript.
> > >
> > > Addressing your ongoing concerns about the scientific value of fine-tuning LLaMA for specialized financial tasks, we would like to elaborate further.
> > >
> > > Firstly, the utility of Multi-Task Learning (MTL) and Supervised Fine-Tuning (SFT) extends beyond financial tasks and across various domains. In this regard, Shah et al. highlight the limitations of zero-shot LLMs and show how SFT can bring about improvements in financial tasks[1]. This parallels our findings and enriches our study's contribution. Further, the role of instruction tuning has been confirmed in a variety of sectors like biomedicine, law, home renovation, e-commerce, education, and writing assistance[2-8].
> > >
> > > Secondly, based on our comprehensive benchmark and the fine-tuning of the LLaMA model on financial tasks, we assert that the direct application of LLaMA in a multi-task setting is insufficient for specialized financial applications. Our empirical findings substantiate this point.
> > >
> > > Thirdly, we have released the first multi-task instructional dataset for SFT in the financial sector, and are the first benchmark that can evaluate fine-tuned models along with backbone models on multiple financial tasks. This resource will serve as a cornerstone for researchers and practitioners interested in applying LLMs in finance.
> > >
> > > We emphasize that our contributions are multi-layered, providing not just new empirical frameworks but also unique data resources and fresh insights into the capabilities and limitations of language models in financial applications.
> > >
> > > Due to space limitations in the manuscript, we intend to offer a more comprehensive comparison in an appendix. Additionally, we are conducting experiments on other tasks within our benchmark, and will report the results upon completion.
> > >
> > > We sincerely hope our response addresses your concerns in a comprehensive manner.
> > >
> > > Sincerely,
> > >
> > > PIXIU Authors
> > >
> > > References:
> > >
> > > [1] Shah, Agam, and Sudheer Chava. "Zero is Not Hero Yet: Benchmarking Zero-Shot Performance of LLMs for Financial Tasks." ArXiv abs/2305.16633 (2023): n. pag.
> > >
> > > [2] Li, Chunyuan, et al. “LLaVA-Med: Training a Large Language-and-Vision Assistant for Biomedicine in One Day.” ArXiv abs/2306.00890 (2023): n. pag.
> > >
> > > [3] Wang, Hao, et al. “HuaTuo: Tuning LLaMA Model with Chinese Medical Knowledge.” ArXiv abs/2304.06975 (2023): n. pag.
> > >
> > > [4] Cui, Jiaxi, et al. “ChatLaw: Open-Source Legal Large Language Model with Integrated External Knowledge Bases.” ArXiv abs/2306.16092 (2023): n. pag.
> > >
> > > [5] Wen, Cheng, et al. “ChatHome: Development and Evaluation of a Domain-Specific Language Model for Home Renovation.” ArXiv abs/2307.15290 (2023): n. pag.
> > >
> > > [6] Li, Y., et al. “EcomGPT: Instruction-tuning Large Language Model with Chain-of-Task Tasks for E-commerce.” ArXiv abs/2308.06966 (2023): n. pag.
> > >
> > > [7] Dan, Yuhao, et al. “EduChat: A Large-Scale Language Model-based Chatbot System for Intelligent Education.” ArXiv abs/2308.02773 (2023): n. pag.
> > >
> > > [8] Zhang, Yue, et al. “Multi-Task Instruction Tuning of LLaMa for Specific Scenarios: A Preliminary Study on Writing Assistance.” ArXiv abs/2305.13225 (2023): n. pag.

---

> > > > ### Comment · Reviewer_v2GB · 2023-08-30
> > > >
> > > > Thanks for the update.
> > > >
> > > > I have a question about "the role of instruction tuning has been confirmed in a variety of sectors like biomedicine, law, home renovation, e-commerce, education, and writing assistance[2-8]." Do these papers have been published? It seems like all these attempts have not been peer-reviewed yet and thus hard to have a demonstration effect.

---

> > > > > ### Author Response · Authors · 2023-08-30
> > > > >
> > > > > Dear Reviewer v2GB,
> > > > >
> > > > > Thank you for your insightful question about the peer-review status of the papers we cited.
> > > > >
> > > > > While it's true that many of these studies are still under review, we'd like to highlight that the LLaMA model itself was submitted in arXiv as recently as February 2023 and has not yet been peer-reviewed. Given the quick pace of research in this area and the duration of the peer-review process, many of these works are likely in the review phase for major conferences such as EMNLP or NeurIPS, which are still in the rebuttal phase.
> > > > >
> > > > > Furthermore, it's important to acknowledge that instruction tuning and multi-task learning have already seen validation in peer-reviewed venues. For example, there are papers published in this year's ACL [1][2][3], and papers in Nature [4] also included instruction fine-tuning. Given these published works, we believe that instruction tuning, especially with emerging Large Language Models like LLaMA, will play an increasingly significant role in both research and application.
> > > > >
> > > > > Sincerely,
> > > > >
> > > > > PIXIU Authors
> > > > >
> > > > > References:
> > > > >
> > > > > [1] Po-Nien Kung and Nanyun Peng. 2023. Do Models Really Learn to Follow Instructions? An Empirical Study of Instruction Tuning. In Proceedings of the 61st Annual Meeting of the Association for Computational Linguistics (Volume 2: Short Papers), pages 1317–1328, Toronto, Canada. Association for Computational Linguistics.
> > > > >
> > > > > [2] Yin, Fan, Jesse Vig, Philippe Laban, Shafiq R. Joty, Caiming Xiong and Chien-Sheng Wu. “Did You Read the Instructions? Rethinking the Effectiveness of Task Definitions in Instruction Learning.” Annual Meeting of the Association for Computational Linguistics (2023).
> > > > >
> > > > > [3] Jingwei Ni, Zhijing Jin, Qian Wang, Mrinmaya Sachan, and Markus Leippold. 2023. "When Does Aggregating Multiple Skills with Multi-Task Learning Work? A Case Study in Financial NLP." In Proceedings of the 61st Annual Meeting of the Association for Computational Linguistics (Volume 1: Long Papers), pages 7465–7488, Toronto, Canada. Association for Computational Linguistics.
> > > > >
> > > > > [4] Singhal, Karan, et al. "Large language models encode clinical knowledge." Nature (2023): 1-9.

---

### Official Review · Reviewer_kysA · 2023-07-22
**Review of the PIXIU benchmark**

**Rating:** 8
**Confidence:** 4

**Strengths:**

The proposed dataset is the first instruction tuning corpus covering the financial domain. Unlike the prior works on the field, both the model and the data are publicly available.
The dataset covers multiple financial NLP tasks and one financial prediction task.
The model released by the author is the first LLM fine-tuned with instruction data and it achieves superior performance in most tasks with respect to other popular non-fine-tuned models. I believe that the public availability of the dataset is the main contribution of the paper, which is strengthen by the related fine-tuned model and evaluation benchmark.

**Additional Feedback:**

1.	Do you have any ideas on how to possibly improve performance on the stock movement prediction task?
2.	How can it be explained that your model achieves lower results compared to baselines in QA, which you state that requires complex numeric reasoning, but better or comparable results in stock movement prediction? Shouldn’t the stock movement prediction task require more advanced numeric reasoning capabilities?

**Clarity:**

Overall, the paper is well written and well organized in sections. A list of modifications and/or requests for clarification that would improve the quality of the work follows.

-	row 14: “five financial NLP tasks” should be “four financial NLP tasks”
-	row 45: According to the sentence, it seems that FinMA is the first financial LLM whereas it is the first according to certain features (public availability of data and model, instruction tuning data format, financial prediction task)
-	row 71: Please define the “FIT” acronym (defined later in the paper)
-	row 79: FinMA does not outperform ChatGPT and GPT-4 on the NER task
-	Table 2: only for the headline classification and NER tasks, dataset names (i.e., Gold, FIN) are not reported. Also later in the paper they are referred to as “Headline” and “NER” datasets (e.g., rows 254, 257, Table 5)
-	row 178: It seems there is a wrong/missing hyperlink
-	row 183: Did you use only one instruction also for Headline (i.e., Gold) and FinQA datasets?
-	Figure 1: the example prompt for BigData22 is different from that in Table 1
-	row 221: “covers financial prediction tasks”, however at the moment there is only one financial prediction task
-	Table 4: typo “Accurady”
-	row 225: in case of unbalanced datasets, it would be more fair to use the macro F1-score
-	row 248: Does “News dataset” refer to the headline classification dataset?
-	Table 5: It would be better to indicate which results are referenced from paper and/or based on human evaluations.
Why are the results of some baselines on some tasks missing?
Vicuna-13B is one of the baselines but its results on all tasks are missing
Results on the FiQA-SA dataset are reported using F1-score while in Table 4 also accuracy is mentioned
-	row 270: typo "FLAPE"
-	Table 6: It would better to include also results from GPTNeo, OPT, BLOOM and ChatGPT
-	row 279: “[LLaMA] presents better performance on mathematical benchmark datasets” but row 261 “[LLaMA] resulted in poor performance on the mathematical benchmark datasets”
-	row 280: also GPT-4 has not been fine-tuned with the financial prediction dataset but it shows better accuracy performance on stock prediction on two datasets with respect to the fine-tuned FinMA-7B-full
-	row 346: There is an error in the BERT reference
-	row 445: There are no error bars, the authors did not specify if results are averaged over multiple runs or not
Supplementary material
-	row 119: Does the task only require verifying if the headline mentions the price? In row 159 of the main paper, 9 different tags were defined
-	row 203 and following: Split sizes of the FPB dataset do not sum to 4845 (Table 2), same for NER dataset (i.e., FIN), FinQA and ConvFinQA. I assume that such split sizes refer to the raw instances (not in the instruction format)
-	row 217: It would be clearer to provide a reference for the previous methods mentioned
-	row 396 and following: It would be beneficial to indicate the dataset to which each reference refers

**Correctness:**

The authors’ claims are generally correct, minor comments or requests for clarification are reported in the “Clarity” section of this review.

The proposed dataset derives from already existing datasets which are assembled to cover multiple financial NLP and prediction tasks. The collection and pre-processing procedures of the dataset are exhaustively detailed in the Appendix.

Some of the results reported by the authors are taken from previous works and/or based on human evaluations. To allow a fair comparison between the baselines considered, it would be better to clearly indicate when results have not been recomputed but they have been taken from their original works.

Moreover, the results of some baselines on some tasks are missing.

**Documentation:**

The authors provide full details on the data collection process, organization, availability, maintenance, potential future updates and hosting of the dataset. They also release the code to run the evaluation performed in their work.

**Ethics:**

Since the dataset proposed by the authors derives from already existing datasets, there should not be ethical concerns. The authors also declare the absence of privacy concerns since the original works that released the datasets should have already addressed potential issues.

**Limitations:**

The main cons is that the dataset is an assembly of already existing datasets.This limits the novelty of the contribution.

**Opportunities For Improvement:**

1.	The authors highlight the importance of financial prediction tasks considering their applicability in real-world scenarios. However, they include in the dataset and related benchmark only one of such tasks (i.e., stock movement prediction). For this reason, it would be beneficial to include at least another financial prediction task.
2.	Regarding the financial NLP tasks, only one out of four is a generative task (i.e., QA). Given the widely recognized performance of LLMs on generative tasks, it would be interesting to consider at least another generative financial task such as the well-known financial text summarization task (which is also included in the BBT-CFLEB benchmark).
3.	The authors experimented with two models fine-tuning them on all four NLP task and one model fine-tuning it on all tasks (4 financial NLP tasks and 1 financial prediction task). It could be interesting to evaluate the performance of a model by fine-tuning it only on the financial prediction task to see how this affects performance.

**Relation To Prior Work:**

The authors clearly discuss the differences between their work and previous contributions. In particular, they focus on the availability of existing datasets and models and highlight the gap in the field they aim at bridging.

**Summary And Contributions:**

The authors present three contributions in the field of financial AI. Specifically, (1) they release an instruction tuning dataset covering four financial NLP tasks and one financial prediction task; (2) they fine-tune LLaMA with the collected instruction data; (3) they propose an evaluation benchmark on the five addressed tasks comparing their fine-tuned LLM with other popular LLMs.

Main pros:it presents the first instruction tuning corpus covering multiple financial NLP tasks.

Main cons: the dataset is an assembly of already existing datasets.This limits the novelty of the contribution.

ADDITIONAL NOTE:
Based on the authors' feedback, I have increasing my rating from 6 to 8 (Clear Accept).

---

> ### Author Response · Authors · 2023-08-22
>
> Dear Reviewer kysA,
>
> Thank you for your comprehensive review and valuable comments on our paper.
>
> **Suggest adding financial predictions and text summarization task:**
> Acknowledging your valuable suggestion, in section 3, we have incorporated one additional financial forecasting task: credit scoring with two datasets, and incorporated two datasets ECTSUM and EDTSUM, for extractive and abstractive financial text summarization respectively. Moreover, we have also added a new dataset for the hawkish-dovish classification task, and a new dataset for the NER task, which has higher quality. It is noticed that all these added new datasets are only used for evaluation and not used for instruction-tuning, which can used for testing the generalization and emerge ability of LLMs. We aim to provide a living project and benchmark, where we plan to continue to update with new tasks and models.
>
> **Dataset merges existing collections**
> We appreciate the contributions of existing collections and wish to underline that we are the first to publish an instruction-tuning dataset and benchmark specifically for developing and evaluating financial LLMs. Our work incorporates public financial NLP datasets and introduces five new financial forecasting datasets, that are not included in existing financial benchmarks. Our resources are publicly available on GitHub and Huggingface, and we hope that they will serve as a foundation for future research and innovation.
>
> **Experimented with multi-task fine-tuning performance:**
> To address this concern, we have added and published a new FinMA-7B-trade model. This model has been exclusively trained on financial forecasting tasks and now available in huggingface. More details are included in the Section 4.
>
> **Need clarity on baseline comparisons:**
> We've made adjustments in Table 5 to clearly indicate which results were taken directly from their original works. Additionally, we've undertaken experiments to provide results for the missing baselines on specific tasks.
>
> **"FinMA is the first financial LLM", according to certain features**
> We have revised the illustration in Line 45, and indicating FinMA is the first open-sourced fine-tuned financial LLMs.
>
> **Other typos and suggestions:**
> Due to the space limit, we cannot answer all your questions here. We have carefully addressed and corrected all the typos and wrong descriptions and appended more illustrations in the revised version of the paper according your suggestions here, which are now marked in red for clarity. We genuinely appreciate your feedback.
>
> **Ideas on improving stock movement prediction:**
> According to our evaluations: 1) Strengthen LLMs' grasp of numbers through specialized fine-tuning, pre-training and larger model sizes. 2) Enhance LLMs' ability to process structured tabular data, like stock histories, for better correlation detection. 3) Increase the input length to consider more than the current 10-day historical data, enhancing prediction accuracy. Addressing these areas should bolster stock movement prediction performance.
>
> **Concern about the performance of our method and ChatGPT on QA vs Stock Prediction:**
> Our model's lower performance in QA is mainly due to its limitations in numerical reasoning, as it's based on the LLaMA architecture. However, stock movement prediction is not just about numerical reasoning; it's a complex task influenced by various qualitative factors. This complexity is evidenced by ChatGPT's good performance in numeric reasoning but poor results in stock prediction. It's worth noting that although our model performs comparably or better than ChatGPT in stock movement prediction, the overall performance in this task for both models is still relatively poor and close to random guessing.
>
> Thank you once again for your valuable comments, and we hope that the revisions will address your concerns satisfactorily.
>
> Best,
>
> PIXIU Authors

---

> > ### Comment · Reviewer_kysA · 2023-08-22
> > **Comment on the authors' feedback**
> >
> > I have appreciated the answers by the PIXIU authors. Now I champion the paper as I deem its contribution valuable for the research community.

---

> > > ### Author Response · Authors · 2023-08-22
> > >
> > > Dear Reviewer kysA,
> > >
> > > We are truly grateful for your thorough review and your updated score. Your constructive feedback was invaluable in refining our paper, and we are pleased to hear that you now champion the paper's contribution to the research community.
> > >
> > > Thank you again for your positive feedback and endorsement of our research.
> > >
> > > Best regards,
> > >
> > > PIXIU Authors

---

### Author Response · Authors · 2023-08-22

Thank you for your valuable comments and suggestions on our manuscript. We appreciate the time and effort you have invested in reviewing our work. We have carefully addressed each of your comments in the revised manuscript.All changes have been marked in red.

After our recent extensions, the framework now includes the first financial LLM fine-tuned on a large dataset of 128K instruction samples and offers a comprehensive evaluation benchmark featuring 8 tasks and 15 datasets, four of which were added following your insightful suggestions—one being a financial forecasting task.

We believe the value of our work is being a pioneering effort in providing the first open-source data for instruction tuning, establishing the first comprehensive evaluation benchmark, creating the first open-source financial LLMs, and critically examining the strengths and weaknesses of LLMs in the financial domain.

We've invested approximately $20,000 in dataset construction, model fine-tuning, and evaluations, and have plans to extend this investment for future model developments. We have open-sourced all our assets, including the model, datasets, and benchmarks.
Our goal is to maintain an evolving project and benchmark that we intend to continually update with new tasks, models, and datasets. Our sincere aspiration is that these endeavors will foster greater transparency and drive progress in the field of financial artificial intelligence.

---

### Decision · Program_Chairs · 2023-09-22

**Decision:**

Accept (Poster)

**Comment:**

Initially, the reviewers had many concerns and questions about this paper. The authors addressed most of these concerns and questions in their responses and the revised version, leading to an improvement in the overall scores from the reviewers. However, there are still remaining concerns regarding the scientific values of fine-tuning LLaMA on a downstream domain and the performance gap between the decoder-only LLM and other models of the selected tasks. Hopefully, the comments and suggestions from reviewers can help the authors further improve this paper.